# Photon-phonon collaboratively pumped laser

Yu Fu[1,3], Fei Liang [1,3], Cheng He [2], Haohai Yu [1]✉, Huaijin Zhang [1]✉ & Yan-Feng Chen [2]✉

In 1917, Einstein considered stimulated photon emission of electron radiation, offering the theoretical foundation for laser, technically achieved in 1960. However, thermal phonons along with heat creation of non-radiative transition, are ineffective, even playing a detrimental role in lasing efficiency. Here, we realize a photon-phonon collaboratively pumped laser enhanced by heat in a counterintuitive way. We observe a laser transition from phonon-free 1064 nm lasing to phonon-pumped 1176 nm lasing in $Nd:YVO_4$ crystal, associated with the phonon-pumped population inversion under high temperatures. Moreover, an additional temperature threshold ($T_{th}$) appears besides the photon-pump power threshold ($P_{th}$), and a two-dimensional lasing phase diagram is verified with a general relation ruled by $P_{th} = C/T_{th}$ (constant C upon loss for a given crystal), similar to Curie's Law. Our strategy will promote the study of laser physics via dimension extension, searching for highly efficient and low-threshold laser devices via this temperature degree of freedom.

A laser is a coherent, ordered, and directional light beam generated by amplification of the stimulated emission of radiation, theoretically originated by Einstein in 1917[1], then technically realized by Maiman in 1960 using a ruby crystal in a resonant cavity[2]. Lasing will spontaneously generate when pump power beyond a threshold ($P_{th}$) overcomes the critical loss. In the past half a century, laser technology has triggered a series of breakthroughs, significantly advancing both fundamental research and our daily life, such as chirped pulse amplification, optical communication, medical treatment, and autonomous driving[3–5]. To date, a vast diversity of laser wavelengths ranging from terahertz to deep-ultraviolet window[6–8] and laser media[9–12], e.g., solid-state, gas, excimer, and semiconductor lasers, have been discovered, thus playing significant roles in particular application scenarios. Specifically, solid-state lasers based on crystals doped with active ions, as the first invented laser type, have been attracting intensive attention from the very beginning due to their high efficiency, low cost, compactness, and long lifetime.

However, solid-state crystal lasers usually suffer from limited available wavelength ranges and strong thermal effects, such as thermal lensing in the gain medium[13], hindering their further

applications. For example, the accessible wavelengths upon intrinsic energy levels of doping ion and the $P_{th}$ upon gain coefficient of materials are already determined for a given laser crystal. The lasing wavelength is usually constrained inside the fluorescence spectrum, switching on or off solely decided by the pump power. Moreover, the useless byproduct heat generated in non-radiative transitions leads to low efficiency, even deteriorated laser performances sometimes needing expensive thermal management to remove it[14]. In previous reports, to the best of our knowledge, almost all solid-state lasers benefited from low temperature with reduced $P_{th}$ and increased slope efficiency, e.g., ruby laser, Ti:sapphire, Yb:YAG, $Nd:YVO_4$. Only a few alexandrite lasers exhibited the reduced $P_{th}$ at high temperatures owing to its special crystal filed splitting and thermally-enhanced $^4T_2$ population[15–18] (A comprehensive summary is listed in Supplementary Fig. S1 and Table S1). As a result, lots of efforts were devoted to growing various kinds of laser crystals with high thermal conductivity or building complicated cooling systems compromisingly. Thus, a solid-state laser with high-temperature tolerance and low $P_{th}$ under heating, preferably a new and tunable degree of freedom, has long been sought.

[1]State Key Laboratory of Crystal Materials and Institute of Crystal Materials, Shandong University, Jinan, China. [2]National Laboratory of Solid State Microstructures & Department of Materials Science and Engineering, Nanjing University, Nanjing, China. [3]These authors contributed equally: Yu Fu, Fei Liang. ✉e-mail: haohaiyu@sdu.edu.cn; huaijinzhang@sdu.edu.cn; yfchen@nju.edu.cn

Back to the quantum theory of the laser, lasing emerges when breaking thermal equilibrium conditions dominant in the spontaneous emission process. In general, a multi-level system was adopted to easily realize population inversion. Thus, besides desired photon emission, there are ineffective but inevitable non-radiative transitions[19], producing heat associated with active lattice vibrations, i.e., phonon. It is well understood these random thermal phonons will perturb intrinsic electronic states involved in the photon emission, leading to fluorescence spectra homogeneous broadening. Recently, our group realized a multiphonon-assisted lasing[20,21] far beyond the inherent fluorescence spectrum in Yb:YCa$_4$O(BO$_3$)$_3$ and Yb:La$_2$CaB$_{10}$O$_{19}$ crystal. However, it is still challenging by directly coupling incoherent phonons to electrons, to make the coherent lasing beyond the inherent fluorescence spectra. Moreover, the relation in the lasing process remains lacking when phonon is coupled to electrons coherently, especially under high temperatures.

In this work, by selecting and amplifying specific-symmetry photon emission, we realize a photon-phonon collaboratively pumped (PPCP) laser enhanced by heat. A clear laser transition from phonon-free 1064 nm to phonon-pumped 1176 nm lasing, is observed in Nd:YVO$_4$ crystals. More surprisingly, besides ordinary $P_{th}$, such a phonon-pumped laser provides us an extra and conveniently controllable degree of freedom that is temperature threshold ($T_{th}$), ruled by a simple relation $P_{th} = C/T_{th}$. The constant C is determined by the given active materials.

## Results

### Photon-phonon collaborative mechanism

We use a four-level Nd$^{3+}$-laser configurational model to describe the so-called collaborative-pumping mechanism (Fig. 1a). In the presence of pumping light (denoted as photon pump), the electrons on the ground state will be excited to the top one, then rapidly relaxed to the lasing upper-level $^4F_{3/2}$ with non-radiative transitions. For the conventional lasing of pure electron radiation ($^4F_{3/2} \rightarrow \, ^4I_{11/2}$), the population inversion happens between 0" and 0' state with $n_{(0'')} > n_{(0')}$, leading to the phonon-free lasing, e.g., zero-phonon line (ZPL) laser at 1064 nm in Nd:YVO$_4$.

However, in a real crystal, both lasing upper and lower levels are coupled to lattice vibrations, and the actual electron population is dependent on the density of phonon states. As the temperature increases, the phonon occupation numbers become larger and larger at 1" and 2' states, and electron can absorb phonon energy and move to high levels. Clearly, this is a thermally-activated phonon-pumping process. When a population inversion is created between 1" and 2' state with $n_{(1'')} > n_{(2')}$, the PPCP laser is possible to happen by applying a rational resonant cavity. We deem this as a phonon pumping process, as shown in Fig. 1a.

To clarify the relationship between the population inversion of PPCP laser and temperature, we solve the relevant rate equation for the steady-state condition (Supplementary Materials, Section-VII). The population inversion $\Delta n$ can be written as Eq. (1),

$$\Delta n = n_{1''} - n_{2'} = \frac{W_p[S_{2'0} - e^{\frac{E_{1''}-E_{0''}}{k_B T}}(W_{0''2'} + A_{0''2'}) - (W_{1''2'} + A_{1''2'})]}{[e^{\frac{E_{1''}-E_{0''}}{k_B T}}(2S_{0''1''} + W_{0''2'} + A_{0''2'}) - (W_{1''2'} + A_{1''2'})]S_{2'0}}$$

(1)

where $n_i$ represents the electron population densities of level i, $W_{ij}$ is the transition probability for stimulated radiation between level i and j, $A_{ij}$ is the spontaneous transition probability, and $S_{ij}$ is the nonradiative transition probability, $W_p$ is optically pump rate, $E_{1''}$- $E_{0''}$ is the energy separation between level 1" and 0", $k_B$ is Boltzmann's constant, T is temperature. It is seen that higher temperature T is helpful for the formation of population inversion. If the temperature is sufficiently high, a two-phonon pumping process can also be expected, corresponding to the upper level of 2" state. Therefore, there is actually a synergetic photon-phonon pumping process. In a circle, electron return back to the ground state with two non-radiative transition processes, 2' → 0' and LS → GS ($^4F_{11/2} \rightarrow \, ^4I_{9/2}$ in Nd$^{3+}$-crystals), associated with thermal phonon emission and heat creation.

This phonon-pumped mechanism can be examined by the fluorescence emission. Besides the ZPL and fluorescence spectral branches,

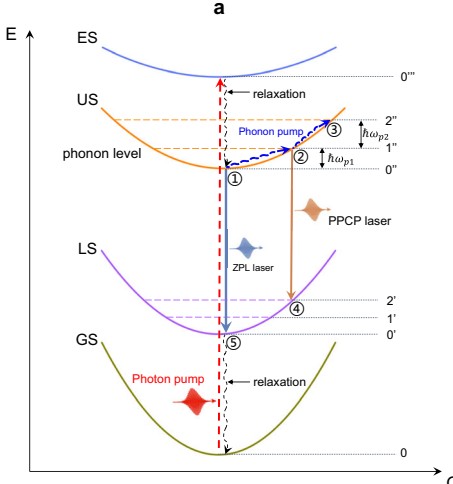

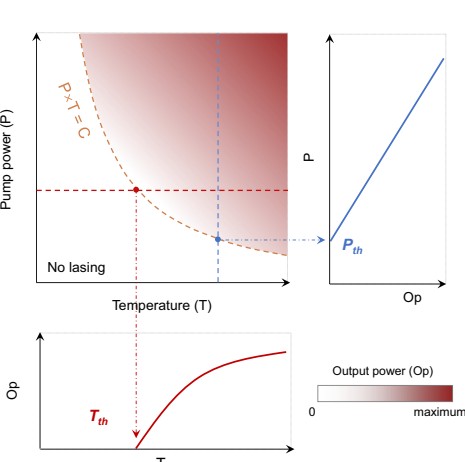

**Fig. 1 | Photon-phonon collaboratively pumped mechanism. a** Configurational coordination model for phonon-pumped lasing. GS ground state, LS laser low-level state, ES excited state, US laser up-level state. The relaxation processes represent the non-radiative transitions. The red dash arrow is a photon-pumping process, and the blue arrow represents a phonon-pumping process. ZPL laser is a phonon-free laser at the zero-phonon line wavelength. PPCP laser is a photon-phonon collaboratively pumped laser. $\hbar\omega$ is the phonon energy. The dash lines are phonon energy levels coupled to a given electronic state. In conventional Nd$^{3+}$-lasers, ①→⑤ transition at 1064 nm is natural at a finite temperature. It represents a phonon-free

laser oscillation. If we suppress the ①→⑤ laser oscillation by specific coating cavity, and amplify another transition channel ②→④, photon-phonon collaboratively pumped lasing at 1176 nm become available. At this time, a population inversion happens between ② and ④ states ($n_② > n_④$ for lasing), when there are enough active ions at ① state and sufficient thermal phonons to support ①→② up-moving. **b** 2D (P, T) lasing phase diagram spanned by temperature and pump power. The color scale represents output power (the color white represents no lasing). The general threshold curve (dashed orange line) satisfies $P_{th} \times T_{th}$ = Constant. Vertical (horizontal) slice is plotted in the right (bottom) panel.

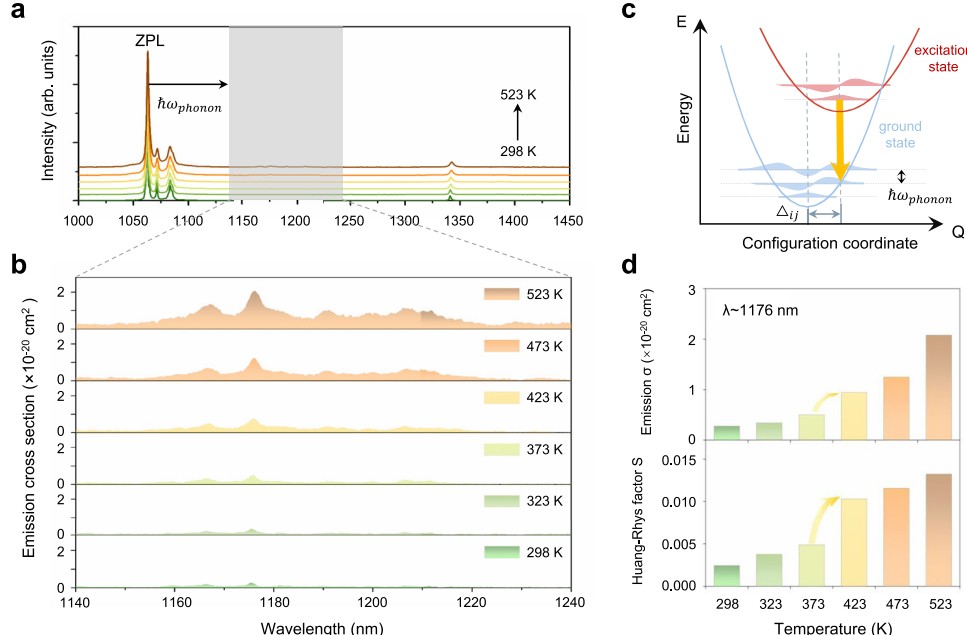

**Fig. 2 | Electron-phonon coupling in fluorescence. a** Fluorescence spectra (π-polarization) at various temperatures. The shadow region is the phonon-triggered emission window. A similar σ-polarization emission is plotted in Fig. S2. **b** Emission cross sections in 1140–1240 nm window. The fluorescence lifetimes of various emission are fitted in Supplementary Fig. S3-S4. **c** Configuration coordinate model.

The forced electronic transition by phonon is represented by the orange arrow, the horizontal dashed lines represent the phonon states, and the vibrational wave functions were plotted as shadow area in the case of quasi-harmonic oscillators. **d** The emission cross-section at 1176 nm and the calculated Huang-Rhys *S* factors at various temperatures.

the electrons on the phonon-pumped levels can induce additional electronic radiation, but this thermal phonon-triggered emission is nearly unobservable, e.g., three orders of magnitude smaller, due to the naturally weak electron-phonon coupling in $Nd^{3+}$-materials[22]. However, strong phonon-pumped lasing still be expected because the resonant cavity can amplify this weak emission. When the ZPLs are prohibited, those electrons on phonon-pumped levels are forced radiation, then creating lasers at new wavelengths. The involved phonons with particular symmetry are spontaneously ordered and significantly enhanced into a coherent type, leading to stronger lattice vibrations and bringing more heat, thus forming a positive feedback loop.

Considering two limiting temperature cases: (1) enough phonons at high temperature, lasing decided solely by photon pumping power, namely $P_{th}$; (2) few phonons at low temperature (~0 K), no lasing happens without enough phonon participation. Therefore, a temperature threshold $T_{th}$ must exist, corresponding to the critical coherent phonon numbers to trigger such phonon-pumped lasers. The actual population at 1″ (or 2″) state is determined by photon-pumping (input power) and phonon-pumping (temperature) simultaneously. Thus, we attempt to draw the lasing phase diagram in a two-dimensional (2D) parameter space spanned by pump power and temperature, as shown in Fig. 1b. Considering such radiation probability relying on the overlap of photon and phonon wave functions, one may thus expect the relation as Eq. (2):

$$P_{th} = C/T_{th}, \qquad (2)$$

where $C$ is a constant, probably upon critical loss of a given laser crystal. In the following, we focus on experimentally verifying the above proposal.

## Electron-phonon coupling in fluorescence

We start with the thermal fluorescence spectra as a basis for our PPCP laser. In experiments, we choose a widely used laser crystal $Nd:YVO_4$ with highly efficient radiation transition of $4f$ electrons in active $Nd^{3+}$

ions[23]. As shown in Fig. 2a, the thermal fluorescence spectra have two conventional fluorescence emission windows originating from electronic transitions, located around 1064 nm and 1342 nm. We mainly focus on the phonon-triggered emission window, i.e., 1160–1220 nm, which shows weak intensity and tiny fluctuations at high temperatures. More explicitly in Fig. 2b, we calculated their emission cross sections, increasing with temperature owing to strengthened electron-phonon coupling. Their over 60 nm bandwidths indicate that it is the phonon frequencies across the entire Brillouin zone, rather than discrete ones, that contribute to the broadband emission in this window.

Here, we use the configuration coordinate model to analyze the electron-phonon coupling and phonon-triggered emission[24] (Fig. 2c). Via Franck-Condon approximation[25], the overlap integral of phonon wave functions between ground and excited states, dominates the electron-phonon transition probability. Electron-phonon coupling makes equilibrium positions of lattice vibration shifting $\triangle_{ij}$[26]. Under harmonic approximation, phonons act as a one-dimensional harmonic oscillator. Thus, $\triangle_{ij}^2 \propto \hbar\omega_{phonon}$, is further proportional to temperature ($T$) $\triangle_{ij}^2 \propto k_B T$. Therefore, electron-phonon coupling strength greatly increases with the increasing temperature.

Based on solid-state physics, the electron-phonon coupling strength could be quantitively evaluated by Huang-Rhys factor ($S$)[19]. The calculated results (Fig. 2d and Fig. S5) show the extremely weak strength with small $S$ factors (0.002–0.013), several orders of magnitude smaller than those in semiconductors (10–100)[27] or transition metal ions doped cases (1–4)[28]. Despite being weak, there is an obvious step in the Huang-Rhys factor as well as emission cross-sections near 400 K, which is related to the thermally strengthened electron–phonon coupling effect. Restoring to aforementioned collaborative-pumping mechanism, this jump indicates the significantly increased phonon numbers involved in the phonon-triggered emission.

## Selecting specific-symmetry phonons

The electron-phonon coupling strength in Fig. 2 is an average interaction of all phonons in the whole window. However, we can see there

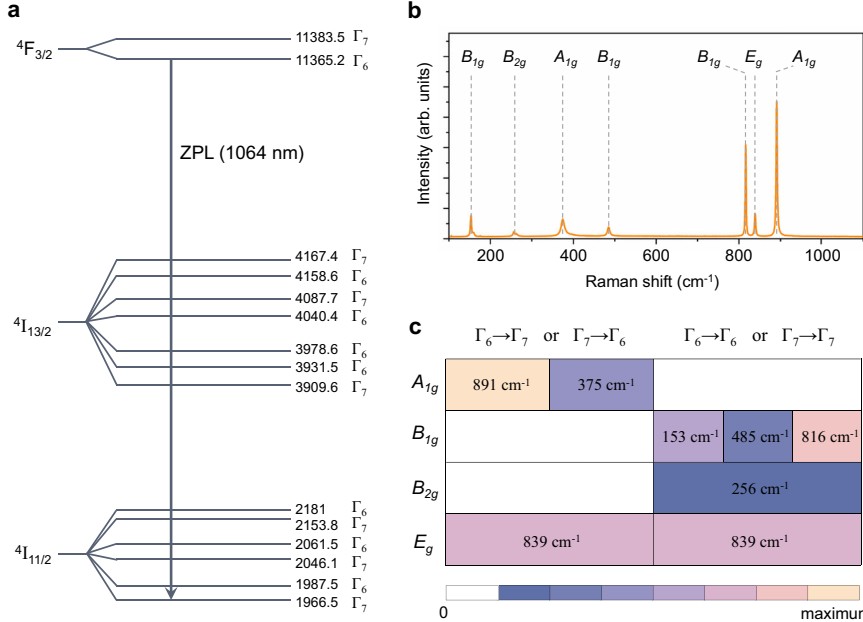

**Fig. 3 | Symmetry requirement of phonons. a** Hyperfine energy levels of Nd:YVO$_4$. Solid lines represent electronic states marked with wavenumber (cm$^{-1}$) and group irreducible representation $\Gamma_6$ or $\Gamma_7$. ZPL represents 1064 nm emission. **b** Raman spectrum marked with phonon modes. **c** Phonon modes allowed in different electron radiation conditions. Insets are phonon wavenumbers. The color scale represents the electron-phonon-coupling intensity.

are several relatively strong peaks at 1168 nm, 1176 nm, 1191 nm, etc., corresponding to some reinforced transitions from phonon-pumped levels. Accordingly, we should further figure out which states of electrons, what modes of phonons could be involved, and how they interplay. YVO$_4$ crystal belongs to tetragonal I4$_1$/amd space group, where the Y$^{3+}$ (doped Nd$^{3+}$) ions in the $D_{2d}$ site reside between (VO$_4$) tetrahedral units[29]. Then, we can obtain the hyperfine electronic energy levels considering the crystal-field splitting effect of doped active Nd$^{3+}$ ions. As shown in Fig. 3a, they split into a series of hyperfine levels with different symmetries denoted as $\Gamma_6$ or $\Gamma_7$[30]. Given the synergetic photon-phonon pumping model above, the expected lasing upper levels and lower levels would appear above $^4F_{3/2}$ and $^4I_{11/2}$, respectively.

Regarding phonons, although we can pre-know all possible modes based on YVO$_4$ crystal structure (neglecting doping influence), not all of them contribute to electron-phonon couplings. Only those modes bringing changed transition polarizability, work in electron-phonon coupling process (Supplementary Materials, Section-V), which can be extracted from the Raman spectrum. As shown in Fig. 3b, seven peaks correspond to four irreducible representations of phonons, i.e., A$_{1g}$, B$_{1g}$, B$_{2g}$, and E$_g$ (D$_{4h}$ point group).

Since different electron radiations may only permit specific-symmetry phonons, their interplay is key to selecting symmetry-allowed phonons. Under an electric dipole moment operator ($\hat{M}$), the transition probability from the initial state $i$ to the final state $f$ is dependent on its matrix element $\langle \psi_f \varphi_f | \hat{M} | \psi_i \varphi_i \rangle$ in Born-Oppenheimer approximation, where $\varphi$ ($\psi$) represents wave function for electrons (phonons). This element is zero unless the symmetries of final wave functions are included in both initial ones and operators or their combinations (Supplementary Materials, Section-V), manifesting as Eq. (3):

$$D\left(\varphi_f\right) \otimes D\left(\hat{M}\right) \otimes D\left(\psi_i\right) \otimes D\left(\varphi_i\right) = D\left(\psi_{f1}\right) \oplus D\left(\psi_{f2}\right) \oplus D\left(\psi_{f3}\right) \oplus \ldots$$

(3)

In our a-cut crystal, $\pi$-polarization emission corresponds to a z-component $\hat{M}$ with B$_2$ group representation. Thus, we can assign the phonon modes allowed to form phonon-pumped levels in different electron radiation conditions (Fig. 3c), by selecting symmetry species of D$_{2d}$ group for Nd$^{3+}$ electronic levels and D$_{4h}$ group for Nd:YVO$_4$ phonons (see Fig. S6). For radiations under the same symmetry ($\Gamma_6 \to \Gamma_6$ or $\Gamma_7 \to \Gamma_7$), B$_{1g}$, B$_{2g}$, and E$_g$ phonon modes are allowed, while under different symmetries ($\Gamma_6 \to \Gamma_7$ or $\Gamma_7 \to \Gamma_6$), only A$_{1g}$ and E$_g$ are allowed. This analysis is consistent with the Raman results. Here, PPCP laser naturally arrives by combining the strongest photon-pumped ZPL and the most phonon-pumped levels. Thus, the most possible PPCP laser will appear at 1176 nm, with 1064 nm ZPL coupled to the strongest A$_{1g}$ phonons at 891 cm$^{-1}$. The next one is 1168 nm, coupled to the second strongest E$_g$ mode at 839 cm$^{-1}$, preferably at a relatively high temperature or adjusting intracavity loss by an optical filter.

**Experimental lasing transition phase diagram**

In lasing experiments (experimental setup see supplementary Fig. S7), our a-cut Nd:YVO$_4$ crystal sample has dimensions of $3 \times 3 \times 7.6$ mm$^3$. To suppress the native fluorescence and amplify the phonon-triggered emission, we coated both crystal front-face and the output coupler, with high transmission in the fluorescence area (1055 nm-1085 nm, and 1320 nm-1360 nm), but high reflection in 1100-1200 nm. As shown in Fig. 4, first, the phonon-free 1064 nm laser is dominated at low temperatures. Then, the output power of 1064 nm becomes deteriorated, and phonon-pumped 1176 nm laser gradually excels it with increasing temperature. When photon-pumped power maintains 7.6 W, there exists a critical transition from 1064 nm to 1176 nm lasing around 293 K. The phonon pumping is related to the temperature $T_{th}$ in the critical region. Below $T_{th}$, only the ZPL 1064 nm laser appears due to insufficient phonon supply. Beyond $T_{th}$, the phonon-pumped laser at 1176 nm starts to appear and become stronger and stronger, associated with the coherent phonons in the system. As expected by Eq. (2), this temperature threshold $T_{th}$ is dependent on the input power. As verified in Fig. S8, the temperature threshold $T_{th}$ will increase to around 308 K with a low photon-pumped power of 7.2 W, because the 1176 nm laser needs more supplied phonons in this case. Moreover, the output power of 1176 nm laser tends to be saturated at a high temperature, and earlier upon the lower input photon power.

By utilizing a V-shaped cavity (Fig. S9), we realized a tunable laser in Nd:YVO$_4$ crystal, corresponding to various phonon modes involved

**a**

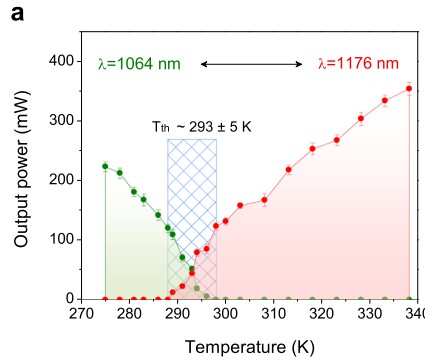

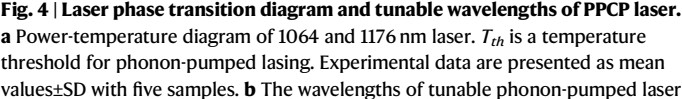

**b**

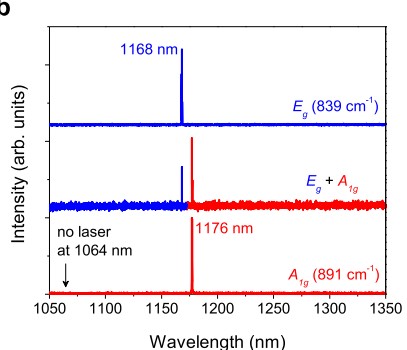

**Fig. 4 | Laser phase transition diagram and tunable wavelengths of PPCP laser. a** Power-temperature diagram of 1064 and 1176 nm laser. $T_{th}$ is a temperature threshold for phonon-pumped lasing. Experimental data are presented as mean values±SD with five samples. **b** The wavelengths of tunable phonon-pumped laser in one-phonon process. Via rotating the birefringent filter (BF) plate, we can obtain the strongest 1176 nm laser coupled to $A_{1g}$ mode, and the second-strongest 1168 nm laser coupled to $E_g$ mode. In addition, these two laser wavelengths can be existed at a special angle of BF plate.

in pumping process. Here, a 1 mm-thick MgF$_2$ birefringent filter (BF) is inserted along the Brewster's angle. By rotating the BF, we realize the lasing wavelength shift from 1176 nm to 1168 nm (Fig. 4b). The participated phonon mode for 1168 nm lasing is 836 cm$^{-1}$ ($E_g$ mode). This is consistent with our proposed selection rules, thus verifying our proposed phonon-pumping mechanism is reliable. Besides one-phonon process, we also observe a two-phonon pumped lasing at 1231 nm (coupled to both 891 and 375 cm$^{-1}$ $A_{1g}$ phonons) (Fig. S10), by using a rational resonant cavity, further verifying the validity of our symmetry analysis. More experimental results of laser performances, including laser linewidth, beam profiles, polarization, and stability, can be found in Supplementary Fig. S11-S31 and Table S2. There is no 1064 nm lasing at high temperature, thus demonstrating 1176 nm and 1168 nm lasers are directly electron-phonon coupling lasers rather than the self-Raman shifted ones.

Finally, we perform 1176 nm laser experiments under different pump power (photon pump) at various temperatures (phonon pump). The lateral-surface temperature of the crystal can be precisely maintained utilizing a thermoelectric cooler (TEC) system. As shown in Fig. 5a, with lateral-surface temperature increasing from 293 K to 338 K, the $P_{th}$ decreased from 7.6 W to 6.6 W while saturation output power increased from 1.5 to 1.7 W, indicating thermally-activated phonon-pumping process indeed exists in Nd:YVO$_4$ crystal. The lowest $P_{th}$ yet highest output power with a slope efficiency $\eta_{slope}$ up to 34.3%, surprisingly occurs without cooling.

Another conclusive evidence for our proposed photon-phonon collaborative mechanism is that the lower temperature (phonon population) makes lower and faster power saturation due to the pumping bottleneck from phonon-pump side. Under low temperature, the population of phonons becomes a shortfall even if the photon-pump power is large enough to make all $A_{1g}$ phonons participate in lasing process. When the temperature is low, phonon number is small, leading to a less contribution to the laser gain in the system. The laser gain mainly depends on the photon pumping in this condition. Therefore, it requires higher photon-pumped power to generate a photon–phonon collaborative pumped laser at low temperatures.

To quantitively investigate such temperature degree of freedom, we need a "real" and physical temperature inside the laser crystal, which is impossible to measure directly. However, we can retrieve the inside thermal-field distribution based on pump power associated with lateral-surface temperature data. Thus, we obtain the characteristic temperature by solving heat transport equation[31], making it possible to study temperature as another degree of freedom in lasing process. As shown in Fig. 5b, with the cooling temperature maintained at 289 K, the maximum temperature value is 461 K (characteristic temperature) under the 7.7 W pump power. This is well consistent with the

temperature threshold $T_{th}$ at approximately 460 K under a fixed pump power 7.7 W. Beyond $T_{th}$, the output power increased with the cooling temperature, first fast, then slow, which is similar to $P_{out}$-$P_{pump}$ relations in Fig. 5c. Thus, like slope efficiency $\eta_{slope}$, we define a thermal slope $T_{slope}$ in our PPCP laser, which is 0.027 W/K in this case. This value represents the increasing rate for involved phonons in PPCP laser. We also find the output nearly reaching saturation over 490 K, indicating the population of phonons is enough while photons turn into a shortfall case. It is expected that the temperature-dependent saturation output power will increase at a higher pump power.

Eventually, we obtain a 2D lasing phase diagram for our PPCP laser depending on crystal temperature and pump power, corresponding to phonon-pump parameter and photon-pump parameter, respectively (Fig. 5d, e). As we expect, the experimental results show a general threshold curve satisfies $P_{th} = C/T_{th}$, with a similar form to the famous Curie's Law[32] (describing magnetic susceptibility of paramagnetic materials). The error bars in Fig. 5d give standard deviation of pump power using the experimental results of five measurements at each cooling temperature. We further obtain the constant $C$ (in the unit W·K) is 3388 with a standard deviation of 158. The tolerance is less than 5%.

## Discussion

Here, we provide a roadmap in Eqs. (4)–(9) to understand how $P_{th}$ and $T_{th}$ connect to each other (see details in Supplementary Materials, Section-VI). Generally, the pump power is[33]

$$P_{pump} = n_{21}\eta\hbar\omega_{photon}/t, \qquad (4)$$

where $n_{21}$ is the population of electrons pumped to the top level, and $\eta$ is the efficiency of non-radiation transition between two excitation levels. Now, we have $n_{21} \propto P_{pump}t$.

On the other hand, in photon-phonon collaborative process, the emission cross section ($\sigma$)[34,35] is proportional to $S$ factor, further proportional to $\triangle_{ij}^2$ as well as $k_BT$

$$\sigma \propto S \propto \triangle_{ij}^2 \propto T/\tau, \qquad (5)$$

where $\tau$ is the fluorescence lifetime. Then, we find parameters related to temperature. As a result, the overall gain ($G$) under small signal approximation in the laser medium can be described as

$$G = n_{21} \times \sigma \propto P_{pump}T \times t/\tau. \qquad (6)$$

When the laser gain overcomes critic loss ($C$), the laser appears. At this time, the fluorescence lifetime $\tau$ is ineffective in a stimulated

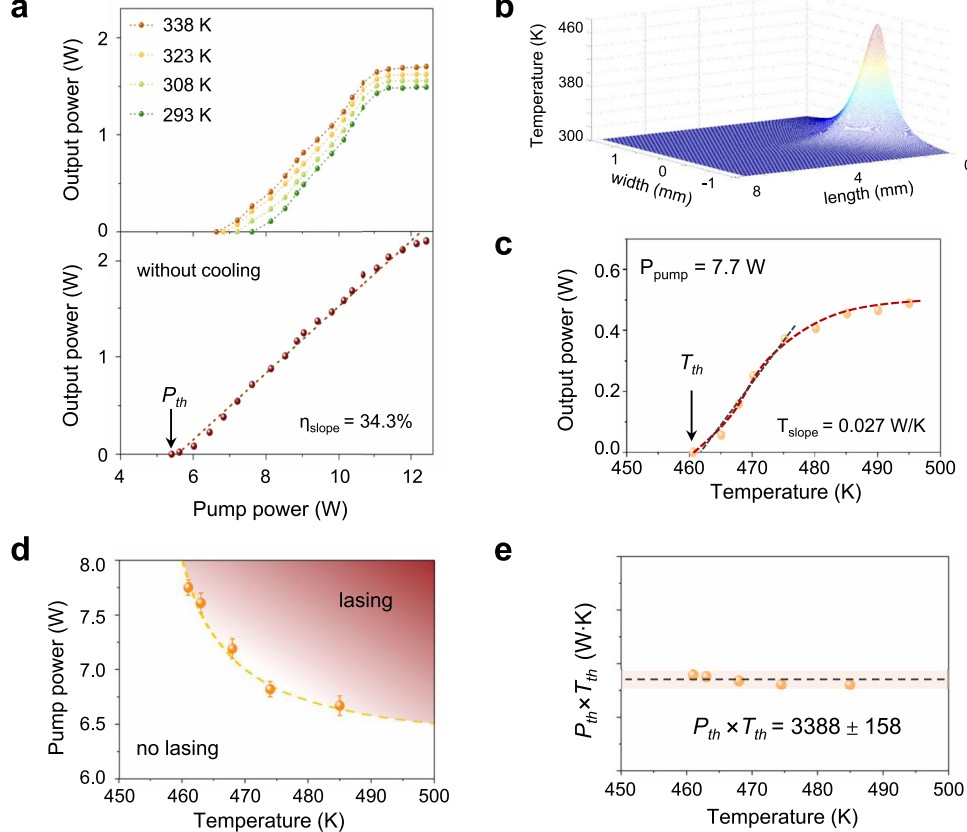

**Fig. 5 | Photon-phonon collaboratively pumped lasing phase diagram.**
**a** 1176 nm laser output power at various cooling temperatures (upper panel) and without cooling (lower panel). **b** Thermal-field distribution inside laser crystal, where the highest value denotes crystal temperature. **c** Crystal temperature-dependent 1176 nm laser output power at a fixed pump power 7.7 W. **d, e** 2D lasing phase diagram and general threshold relation $P_{th} \times T_{th} = C$. The dots are measured results. Experimental data in Fig. 5d are presented as mean values±SD with five samples.

radiation (lasing) process, only providing a time unit to normalize that in the power unit. The general thresholds are exactly at the crucial points when the gain balances the loss. Substituting $P_{pump}$ by $P_{th}$ and T by $T_{th}$, we can get the relation for the general threshold

$$P_{th} \times T_{th} \propto C. \tag{7}$$

For a given laser crystal and resonant cavity, the critical loss is assumed to be a constant, leading the proportional symbol to be equal. In our experiments, this value of $C$ is 3388 ± 158 W·K.

Considering the curve of temperature-dependent output power (Fig. 5c) seems inversely proportional to $T$, we may propose an effective equation for the output power before saturation

$$P_{out} = \eta_{slope} (P_{pump} - C/T). \tag{8}$$

It should be noticed that constant $C$ assumed just on the curve of general thresholds could be different before or after lasing, which also may lead to another critical exponent close to the phase transition. Under the uncooling conditions, the $T$ is dependent on the pump power as $T \propto \alpha \cdot P_{pump}/K_c$ with the absorption coefficient of $\alpha$ and thermal conductivity of $K_c$[34]. Therefore, the output power becomes pump power dependent

$$P_{out} = \eta_{slope} (P_{pump} - A/P_{pump}). \tag{9}$$

This value of $A$ in the present experiments is 22.39 W². Equation (9) considers only the contribution to output power originating from photon pumping, while Eq. (8) takes both photon-pumping and phonon-pumping into account.

Actually, high slope efficiency of 34.3% of our PPCP laser (Fig. 5a) is attributed to the additional heat contribution of approximately 10%. This high efficiency is comparable to diode-pumped self-Raman lasers, optically-pumped Raman lasers, as well as optically-pumped semi-conductor lasers (see Supplementary Materials, Section-VIII, Table S6). After obtaining the general threshold relation, we can assess the $T_{th}$ to be 622 K, without cooling. We can use Eq. (8) to evaluate the temperature-dependent output power under an arbitrary pump power (Fig. 5c), inversely proportional to T with an upshift relying on $P_{pump}$. We note that our lasing phase diagram (Fig. 5d) has its own physical limitation, i.e., the upper limits of $P_{th}$ and $T_{th}$ depend on the saturation of gain and the temperature tolerance of crystal, respectively, and vice versa for their lower limits. It is worthy to note that our PPCP laser does not absorb heat (solid cooling), since those phonon-pumped electrons will finally relax to the ground state by non-radiative transitions, returning heat. Compared to the previous 1176 nm Raman lasers experiencing releasing phonons process that benefits from cooling[36–38], the hallmark of our PPCP one is the existence of $T_{th}$ liking heat; thus, they have opposite causality. Such discussion is mainly applicable to the four-level laser systems. It may become more complex for three-level or other systems, where the formation of population inversion is easily disturbed by high temperatures.

In addition, we need to emphasize that the mechanism of our PPCP laser is fundamentally different from that of alexandrite one[15,16]. Alexandrite laser is a particular case in $Cr^{3+}$-doped laser materials, possessing the distinctive electronic upper levels where two close $^4T_2$ and $^2E$ upper levels are generated by crystal-field Stark splitting. Its laser temperature

enhancement originates from the temperature-dependent Boltzmann distribution in these two levels, whose population can be well described by traditional thermodynamic and statistical physics. However, in our Nd:YVO$_4$ laser, these newly created laser wavelengths were associated with the upper levels induced by electron-phonon coupling on the cost of phonons. The higher temperature will provide more phonons, who acts as a pump source. We give a comprehensive analysis (Supplementary Materials, Section-IV, Fig. S37-S39, Table S5) to demonstrate that phonon-pumping mechanism is a significant step forward in laser physics and laser technology.

To conclude, we realize a PPCP laser, where lasing threshold satisfies a classical and profound formula, $P_{th} = C/T_{th}$. Significantly, a traditional yet newly introduced degree of freedom, i.e., temperature, is a breakthrough in laser physics and technology. Our highly-efficient laser becomes strong by heat, successfully addressing the long-standing restrictions of heat production in lasing process. This collaborative lasing mechanism is promising for searching new-wavelength lasers via phonon symmetry design or multi-phonon coupling processes. The lasing phase diagram we proposed here helps optimize the most efficient routine, maybe need external heating sometimes, probably via machine learning. Our findings also shed light on various fields of cutting-edge and fundamental study, e.g., photonics, optoelectronics and lasing phononics[39–41]. For instance, a thermally-enhanced yellow laser at 588 nm with frequency-doubling technique is obtained in Nd:YVO$_4$, which can apply to retina repair therapy. More advanced applications triggered by such a PPCP laser are still on the way in many scientific fields. Our PPCP laser platform is ready for applications in many scenarios such as high-temperature tolerant laser equipment[42], quantum nanophotonic[43,44], entangled phonons generation[45], and laser diagnosis with new wavelengths.

## Methods

### Experimental setup

Our experimental laser samples are a-cut Nd$^{3+}$-doped YVO$_4$ crystals (Nd$^{3+}$ concentration of 0.46 at.%.) with dimensions of $7.6 \times 3 \times 3$ mm$^3$. The crystal device photographs are plotted in Fig. S7. To remove the extra loss of the input flat mirror and simplify experimental device, we directly coated a high-quality resonant cavity on crystal surfaces. The front surface of crystal is coated with high-reflection (HR, R > 99.9%) at 1176 nm, high-transmission (HT, T > 99.5%) at 1064 nm and 1342 nm, and HT (T > 90%) at 808 nm, 914 nm and 1085 nm, in which the lasing at 914 nm, 1064 nm, 1085 nm, and 1342 nm were completely suppressed. The end surface is coated with anti-reflection (AR, R < 0.1%) at 1176 nm. An output coupler with a curvature radius of 50 mm was employed, whose front surface was coated with HT at 1055 nm-1085 nm, 1320 nm-1360 nm and 1176 nm. The end surface was coated with HT at 1055 nm-1085 nm and 1320 nm-1360 nm, partial reflection (PR) at 1176 nm with the transmission of 1% ± 0.5%.

Moreover, we removed the output coupler to further optimize the resonant cavity. The monolithic configuration was employed for laser generation. An a-cut 0.15 at.% Nd:YVO$_4$ crystal with dimension of $4 \times 4 \times 10$ mm$^3$ was utilized. The front surface was coated with HT at 1064 nm&1083 nm&1342 nm&808 nm, HR at 1176 nm. The end face was HT-coated at 1064 nm&1083 nm&1342 nm and HR-coated at 808 nm&1176 nm.

A commercial laser-diode (LD), with center emission wavelength at 808 nm was adopted as the pump source. The maximum incident pump power reaches up to 25 W. The diameter of the fiber is 400 μm, the pump light from the LD was focused into sample by an imaging unit with a beam compression ratio of 1:1. The pump power refers absorbed value through the whole paper.

The comprehensive laser performances of Nd:YVO$_4$ are summarized here. First, laser experiments exhibit obvious lasing threshold and linewidth narrowing in these crystal devices. The full width at half maximum of the laser lines of a-cut and c-cut Nd:YVO$_4$ are plotted in

Fig. S11 and S12, which are 0.28 nm and 0.12 nm, respectively. The emission linewidths of fluorescence emission (laser emission) are examined below and beyond threshold power (Fig. S13-S17). The linewidths suddenly become narrower when the pump power reaches the threshold at various cooling temperature, providing a direct evidence for the occurrence of PPCP lasing in Nd:YVO$_4$ crystal. In addition, this PPCP laser is linear-polarized and the normalized polarization data was given in Fig. S18, with a degree of polarization (ρ) is 99.97%. This result indicates our laser is a near-perfect linear-polarized light. The beam profiles at various temperatures are depicted in Fig. S19, with the nearly circular symmetric laser beams. The beam quality and beam divergences at 293, 308, 323, and 338 K are calculated in Fig. S20- Fig. S24. All of these results are measured while the output power maintains 1 W. We can see from Table S2, for the crystal with a dimension of $3 \times 3 \times 7.6$ mm$^3$ and the cavity with a $R_{oc} = 50$ mm output coupler, the largest $M_x^2$ and $M_y^2$ are 2.35 and 3.58, respectively. For other configurations, such as different crystal dimensions and cavity lengths, they have no significant impact on the beam quality of PPCP laser (Fig. S25-S26). Moreover, the oscilloscope trance exhibits this PPCP laser is a continuous-wave laser (Fig. S27). In addition, the output power stability of the PPCP laser is examined at an output power of 1.2 W. Fig. S28 shows a curve of output power as a function of time over a period of 30 minutes. The fluctuations of output power were less than ±1.5%. Finally, we also checked the repeatability of PPCP laser at 1176 nm with five different samples. The statistical laser results are listed in Fig. S29 and Table S4.

The laser wavelength shift at various temperatures is displayed in Fig. S30. The ZPL wavelength shifts from 1064 to 1064.25 nm with increasing temperature from 20 to 65 °C. Moreover, the phonon wavenumber of A$_{1g}$ mode shifts from 890.86 cm$^{-1}$ to 890.46 cm$^{-1}$ with increasing temperature from 296 to 338 K (Fig. S31). Accordingly, the phonon-pumped lasing wavelength slightly should shift from 1175.47 to 1175.66 nm. This agrees well with our experimental results if considering measurement errors. In addition, in Fig. S32-S35, we also precluded some alternative explanations for the temperature-dependent characteristics of our PPCP laser, including Raman scattering, pumping absorption, and modification by coating. The absorbed pump power gradually decreased with the increasing temperatures, indicating the low-threshold of 1176 nm laser at high temperature cannot be attributed to the increased pump absorption cross-section. In addition, the thermal effect of traditional Nd:YVO$_4$ laser at 1064 nm was discussed in Supplementary Materials, Section-III, Fig. S36.

### Temperature distribution inside laser crystal

In lasing experiments, the crystal is wrapped with indium foil then mounted in a copper heat sink with temperature controlled by thermoelectric cooler (TEC). The maximum temperature tuning range is from 293 K to 338 K. Notably, this temperature is not the accurate temperature inside Nd:YVO$_4$ crystal. To obtain the temperature inside the laser crystal, we resort to a general form of the steady-state thermal-field distribution equation[46],

$$k_x \frac{\partial^2 T}{\partial x^2} + k_y \frac{\partial^2 T}{\partial y^2} + k_z \frac{\partial^2 T}{\partial z^2} + q(x,y,z) = 0, \qquad (10)$$

where $k_x$, $k_y$, $k_z$ represent the thermal conductivity of Nd:YVO4 crystal alone $x$, $y$, and $z$ directions, respectively. $T(x, y, z)$ is the temperature inside the laser crystal, and $q(x, y, z)$ is the heat generation arising from pumping light per unit volume. The intensity of pumping light is assumed to obey the Gaussian function. The heat generation for Nd:YVO4 crystal on the Cartesian coordinate system can be written as

$$q(x,y,z) = \frac{2Q\alpha}{\pi\omega_p^2}\left(1 - e^{-\alpha l}\right)e^{-2\left[\left(x-\frac{a}{2}\right)^2 + \left(y-\frac{b}{2}\right)^2\right]/\omega_p^2}e^{-\alpha z}, \qquad (11)$$

where $Q$ is the total heat load in the crystal due to the quantum defect, $\alpha$ denotes the pump absorption coefficient, $\omega_p$ is the beam waist, and $l$ is the crystal length.

The Eq. (11) is solved by finite element analysis. The boundary conditions are chosen as the lateral-surface temperature of laser crystal controlled by TEC. Then, we can obtain the real thermal-field distribution inside the crystal.

## Emission cross section calculation

The stimulated emission cross section is essential to evaluate the performance of the laser gain medium. The crystal sample, detector angle, and collection efficiency maintain unchanged in experiments. Only the crystal temperature was changed. With the measured thermal fluorescence spectra (Fig. 2a) and the lifetimes of the upper laser level (Fig. S3), the polarized emission cross sections can be obtained via Fuchtbauer-Ladenburg formula[34,35],

$$\sigma_{em}(\lambda) = \frac{\lambda^4 I(\lambda)}{8\pi c n^2 \tau \int I(\lambda) d\lambda},  \quad (12)$$

where $I(\lambda)$ is the spectral intensity at wavelength $\lambda$, $c$ represents the velocity of light, and $n$ is the refractive index, $\tau$ denotes the fluorescence lifetime of the upper laser level. The refractive index of $Nd:YVO_4$ for $\pi$ polarization can be calculated by Sellmeier equation[47].

## Huang-Rhys $S$ factor calculation

The $S$ factors are calculated based on Huang-Rhys theory under the present condition where the initial phonon number participating the process is assumed to be zero[24],

$$e^{-s} = \frac{I_{ZPL}}{I}  \quad (13)$$

where $I_{ZPL}$ and $I$ are the fluorescence intensities of pure electronic transition and the sum of with and without phonon participation processes, respectively. The intensities of ZPLs and their phonon-triggered emissions can be identified by fitting the peaks of fluorescence spectra of $Nd:YVO_4$ crystal.

## Reporting summary

Further information on research design is available in the Nature Portfolio Reporting Summary linked to this article.

## Data availability

Source data are provided as a Source Data file. Source data are provided in this paper. The data that support the plots within this paper and other finding of this study, are available from the corresponding authors upon request. Source data are provided with this paper.

## Code availability

The code used to calculate the results for this work is available from the corresponding authors upon request.

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

## Acknowledgements
This work was supported by the National Natural Science Foundation of China (NSFC) under Grant Nos. 92163207, 52025021, 51890863, 52372010; National Key Research and Development Program of China Nos. 2021YFB3601504 and 2023YFF0718801.

## Author contributions
H.H.Y. and H.J.Z. conceived and supervised the project. Y.F. and F.L. performed the laser experiments and wrote the manuscript. Y.-F.C. and C.H. provide many helpful suggestions and theoretical analysis for this work. All authors contributed for the discussion and preparation of the manuscript.

## Competing interests
The authors declare no competing interests.
