## [Peer Review File · Nature Communications]

REVIEWER COMMENTS

Reviewer #1 (Remarks to the Author):

In this paper, a phonon-photon collaboratively laser with a lower lasing power threshold P_{th} at higher temperature T_{th} has been reported ($P_{th} \times T_{th} = \text{constant}$), and the underlying mechanism and experimental characterization have been discussed in detail. The experimental results and the related analyses are quite solid and convincing, and this work might pave the way for the future development of laser sources under high temperature. Nevertheless, some important discussions and details are still missing in this manuscript:

1. In my opinion, the author should clarify the relationship between this manuscript and their recent work which they didn't cite here in the introduction part. For instance, Liang, F., He, C., Lu, D. et al. Multiphonon-assisted lasing beyond the fluorescence spectrum. *Nat. Phys.* 18, 1312–1316 (2022). Rulin Miao, Yu Fu, Dazhi Lu, et al, Deciphering the vibronic lasing performances in an electron-phonon-photon coupling system. *Opt. Express* 31, 9790-9799 (2023). Cheng, Y., Liang, F., Lu, D. et al. Phonon engineering in Yb:La₂CaB₁₀O₁₉ crystal for extended lasing beyond the fluorescence spectrum. *Light Sci Appl* 12, 203 (2023).

As the innovation of this paper is about the decreasing lasing threshold with the increased temperature, I think the title “photon-phonon collaboratively pump laser” is not proper, and it is not easy to distinguish this one from the work mentioned above from the title, especially *Nat. Phys.* 18, 1312–1316 (2022). Moreover, even though the author emphasized the distinction between this work and the alexandrite laser, it seems that the underlying mechanism is the same and can be unified, as the theoretical frame and the experimental results about laser threshold versus temperature in *Opt. Express* 31, 9790-9799 (2023) is quite similar to this work. In addition, the comparison of this manuscript and the above work should be added in Figure S1.

2. In the introduction part, lines 64-70 seem to be redundant and it implies some innovation points of this manuscript that are not convincing.

3. The definition of the temperature T_{th} is inconsistent in this manuscript, in figure 4a it is the cooling temperature (293 K) while in figure 5d and 5e it is the crystal temperature (460K-500K).

4. Is the experimental result $C = 3388 \pm 158 \text{ W} \cdot \text{K}$ obtained by fitting the experimental data in Figure 5d of the main text, or by using the theoretical formula with other experiment parameters like optical quality factor? If the answer was the former one, did the author compare it with the theoretical result in the second case?

5. Lines 254-256 “with lateral-surface temperature increasing from 293 K to 338 K, the P_{th} decreased from 7.6 W to 6.6 W while saturation output power increased from 1.5 to 1.7 W which has never been reported to our knowledge” is an ambiguous statement and needs to be revised.

6. It is better to replace the statement “see supplementary material” in the main text (for example, lines 94, 190, 204, and 351) with specific places in supplementary material like “see supplementary material Fig S5”.

7. The citation should be added in line 303 and line 521 (Fuchrbauer-Ladenburg formula).

8. Line 173-174 “When we further amplify it in the resonant cavity, such temperature step could possibly become a lasing threshold T_{th} ” is obscure and needs to be revised.

9. In order to show the system stability and laser noise, the time domain signal shown in Figure S27 is insufficient, the noise represented in the frequency domain by using the electronic spectrum analyzer might be better to characterize the intensity and phase noise of the device here, and further verified the high stability the author announced here.

10. I suggest labeling the variation of the pump power in Figures S14-S17, just like the temperature change elucidated in Figure S30.

11. The text in Figures S32 and S35c is too small.

Reviewer #2 (Remarks to the Author):

The authors demonstrate a lasing phenomenon, which is in general counter to other lasers in the sense that there exist a regime of temperature where the threshold current decreases with increasing temperature. The work is original as far as I can see.

However, the explanations as to the exactly mechanisms of the counterintuitive phenomenon are rather inadequate, in part because of 1) the structure of presentations and 2) language issues. But, setting these stylistic problems aside, I am not certain if the phenomenon is mainly because of increased emission cross-section at higher temperatures, or due to phonon pumping as the title and conclusions suggest. And these two things (cross-section and phonon pumping) seem to be unrelated to each other. In the bulk gain medium, phonon modes have a continue spectrum. Out of all the phonons, what is the percentage of those special phonons that satisfy the symmetric condition for enhanced cross section? Phonon pumping does not seem to increase the population inversion and the total number of ions in the upper lasing state, which is determined by the optical pumping level (phonon only move them to higher energies), how can the output laser power increase with temperature? If the existence of phonons does not increase the number of ions in the upper lasing state, how can you call this phonon pumping? Also, the definition of threshold generally is related to achieving a certain level of population or population inversion. The use of the term “temperature threshold” does not appear to be appropriate if indeed the phonons do not increase population/population inversion. Lastly, in view of the comment above regarding percentage of phonons that satisfy symmetric requirement, what are the power efficiency (accounting for both optical pump power and power used to heat the crystal), in comparison with pure

optical pumping? If there is a penalty in efficiency, what can be the possible applications of this type of lasers.

Reviewer #3 (Remarks to the Author):

Manuscript Report: "Photon-phonon collaboratively pumped laser"

In this manuscript, the authors report on the experimental demonstration of a phonon-assisted (thermally-induced) laser at 1176nm wavelength.

The authors demonstrate a novel laser system that is collectively pumped by both photons and phonons. The authors provide convincing experimental evidence showing conventional power thresholds and input-output power corresponding to a Nd:YVO₄ laser at 1176 nm wavelength. The developed laser is characterized by an unusual temperature lasing threshold: a critical temperature where the output laser increases as the temperature increases while the pump power is sustained. Furthermore, the authors provide a detailed analysis of the temperature dependence which will be very useful to implement laser systems using the different transitions of Nd:YVO₄. To the best of my knowledge, this laser mechanism and the enhanced performance with temperature has never been observed in lasers based on Nd materials. Hence, the novelty of the present work unquestionable and, in my opinion, this work is a valuable contribution for the laser physics community.

I only have a minor concern, I noticed that the authors have published a paper on a similar topic, "Multiphonon-assisted lasing beyond the fluorescence spectrum", in Nature Physics 18, 1312-1316 (2022), and I am wondering how is that work related to the present one? In other words, if both papers deal with phonon-assisted lasers why the nature physics paper is not even cited in the present manuscript?

After clarification of this minor concern I will be in position to recommend the present manuscript for publication in Nature Communications.

Response to reviewer's comments

Reviewer #1

Comment:

In this paper, a phonon-photon collaboratively laser with a lower lasing power threshold P_{th} at higher temperature T_{th} has been reported ($P_{th} \times T_{th} = \text{constant}$), and the underlying mechanism and experimental characterization have been discussed in detail. The experimental results and the related analyses are quite solid and convincing, and this work might pave the way for the future development of laser sources under high temperature. Nevertheless, some important discussions and details are still missing in this manuscript:

*1. In my opinion, the author should clarify the relationship between this manuscript and their recent work which they didn't cite here in the introduction part. For instance, Liang, F., He, C., Lu, D. et al. Multiphonon-assisted lasing beyond the fluorescence spectrum. *Nat. Phys.* 18, 1312-1316 (2022). Rulin Miao, Yu Fu, Dazhi Lu, et al, Deciphering the vibronic lasing performances in an electron-phonon-photon coupling system. *Opt. Express* 31, 9790-9799 (2023). Cheng, Y., Liang, F., Lu, D. et al. Phonon engineering in $\text{Yb:La}_2\text{CaB}_{10}\text{O}_{19}$ crystal for extended lasing beyond the fluorescence spectrum. *Light Sci Appl* 12, 203 (2023).*

*As the innovation of this paper is about the decreasing lasing threshold with the increased temperature, I think the title "photon-phonon collaboratively pump laser" is not proper, and it is not easy to distinguish this one from the work mentioned above from the title, especially *Nat. Phys.* 18, 1312-1316 (2022). Moreover, even though the author emphasized the distinction between this work and the alexandrite laser, it seems that the underlying mechanism is the same and can be unified, as the theoretical frame and the experimental results about laser threshold versus temperature in *Opt. Express* 31, 9790-9799 (2023) is quite similar to this work.*

In addition, the comparison of this manuscript and the above work should be added in Figure S1.

Response:

Thank you for your comments.

(1) First, we cited our three papers in the revision. We first submitted this Nd:YVO₄ work to *Nature Photonics* in 18th-May-2022, and then transferred to *Nature Communications* in 5th-July-2023. At the first submission, these three papers mentioned in the comments have not been published. So, we missed these citations. Now, we added refs [18, 20, 21] in the revision.

(2) Difference from Nd:YVO₄ laser and Yb:YCOB laser in *Nat. Phys.* 18, 1312-1316 (2022).

Nd:YVO₄ laser and Yb:YCOB laser hold different physical mechanism. The former one is the thermally-activated phonon-pumping process with absorbing one or two phonons, whereas the latter one exhibits the multiphonon-assisted laser wavelength extension with emitting three- to eight-phonons.

In previous work in nature physics, we realized a broadband tunable laser generation far beyond the fluorescence spectrum in Yb:YCOB crystal. Yb:YCOB is a strong electron-phonon coupling crystal with Huang-Rhys S factor of 1.34. The main highlight is the great wavelength tuneability and extension by multiphonon-assisted lasing process. We obtained an overall spectral tuneability of 1110-1465 nm in five segments, each with its own configuration, corresponding to the three- to seven-phonon cases. The longest (eight-phonon) lasing wavelength can reach 1518 nm, over 400 nm beyond the fluorescence spectrum.

In this work, Nd:YVO₄ is a weak-coupling system with Huang-Rhys S factor of 0.013. Despite of such small coupling intensity, electron-phonon coupling effect is also important for phonon-involved lasers. Here, we focus on

the temperature-dependent laser performances of Nd:YVO₄ crystal. The main highlight is an anomalous phonon-pumped lasing phenomenon in Nd-laser materials, which is counter to other lasers in the sense that there exists a regime of temperature where the lasing threshold decreases with increasing temperature. A phonon-photon collaboratively pumped laser with a lower lasing power threshold P_{th} at higher temperature T_{th} has been reported ($P_{th} \times T_{th} = \text{constant}$). A critical temperature threshold T_c was verified experimentally to claim the distinctive phonon-pumping mechanism.

(3) Difference from Nd:YVO₄ laser and alexandrite laser.

The physical mechanism of Nd:YVO₄ laser and alexandrite laser is different and can not be unified. The former one is a phonon-pumped lasing process by absorbing symmetry-striction phonons with selection rules. It exhibits a distinct relation, $P_{th} \times T_{th} = \text{constant}$. In comparison, alexandrite laser is a phonon-assisted vibronic lasing process and thermal phonons mainly contribute to spectral broadening. Its temperature-dependent laser threshold don't has such a relation, $P_{th} \times T_{th} = \text{constant}$.

1. The role of phonons

Alexandrite crystal has a large S factor, and the thermal phonons participated in the lasing process play an average effect for *homogeneously spectral broadening* [$S \sim 2.13$ & $\hbar\omega_{ph} \sim 524 \text{ cm}^{-1}$ in our paper *Opt. Express* 31, 9790 (2023)]. Therefore, with temperature increasing from 10 to 50 °C, the laser wavelength has a significant redshift from 761 to 772 nm. The laser wavelength is challenging to keep unchanged, not to mention controllable phonon modes.

Our Nd:YVO₄ crystal has a tiny S factor (0.002-0.013). Only specific-symmetry phonons that follow selection rules, such as A_{1g} or E_g phonon mode coupled to $\Gamma_6 \rightarrow \Gamma_7$ transition, are amplified to *create quantized laser upper-levels*. Thus, we can precisely design and specify laser wavelength maintaining at 1176 nm (coupled to A_{1g} mode) or 1168 nm (coupled to E_g mode). From this perspective, our study may offer an ideal platform to quantitatively verify and study the quantized electron-phonon coupling effect.

Figure R1. Different lasing mechanisms. (a) Alexandrite laser Cr³⁺:BeAl₂O₄, (b) Nd:YVO₄ laser.

2. Different lasing mechanism

Alexandrite laser is only a particular case in Cr^{3+} -doped laser materials, possessing very distinctive electronic levels where two close ${}^4\text{T}_2$ and ${}^2\text{E}$ upper levels are generated by crystal-field Stark splitting (Fig. R1a). Its laser temperature enhancement originates from the temperature-dependent Boltzmann distribution in these two levels, whose population can be well described by traditional thermodynamic and statistical physics. Benefitting from a long lifetime of ${}^2\text{E}$ level, thermal excitation rates for the ${}^2\text{E} \rightarrow {}^4\text{T}_2$ transition increase at high temperature, thus providing enhanced gain on the ${}^4\text{T}_2 \rightarrow {}^4\text{A}_2$ laser transitions.

In our Nd:YVO_4 laser, there is no ${}^2\text{E}$ -like storage level with long lifetime (Fig. R1b). The newly created laser wavelengths associated with the upper levels owe entirely to the enhancement of electron-phonon coupling on the cost of phonons by simultaneously suppressing strong fluorescence emission. The higher temperature will provide more phonon states, who acts as a pump source.

3. Different temperature-dependent relation

In alexandrite lasers, high temperature also tends to populate the terminal ${}^4\text{A}_2$ levels. Therefore, the laser performance is the best one with a maximally populated initial level and a minimally terminal one at an optimal temperature (T_m) to balance the above two conflicting effects. The typical temperature-dependent performance is V-shape [see *IEEE J. Quantum Electron.* 16, 1302 (1980), $T_m \sim 42^\circ\text{C}$ and *Opt. Lett.* 7, 608 (1982), $T_m \sim 225^\circ\text{C}$].

In our Nd:YVO_4 lasers, higher temperature brings more phonons, which will continuously improve laser performances in principle. Most importantly, besides qualitative explanation, we quantitatively obtain a lasing threshold-temperature relation $P_{th} = C/T_{th}$, based on both theoretical analysis and experimental results (C is a constant depending on active material).

(4) Updated Figure S1.

The temperature-dependent laser thresholds of Yb:YCOB , Yb:LCB , and alexandrite laser (circle points), were added in the Fig. S1 and Table S1.

Figure R2. A summary of temperature dependent lasing threshold in solid-state lasers.

Action:

In the revision, we added refs [18, 20, 21] and the revised Fig. S1. The difference between alexandrite laser and Nd:YVO₄ laser was discussed in Supplementary Materials Section-IV.

Comment:

2. In the introduction part, lines 64-70 seem to be redundant and it implies some innovation points of this manuscript that are not convincing.

Response:

Thank you for your comments. We corrected line 64-70 in the revision.

Action:

Line 64-70: “However, it is still challenging by directly coupling incoherent phonons to electrons, to make the coherent lasing beyond the inherent fluorescence spectra. Moreover, the relation in the lasing process remains lacking when phonon is coupled to electrons coherently, especially under high temperatures.”

Comment:

3. The definition of the temperature T_{th} is inconsistent in this manuscript, in figure 4a it is the cooling temperature (293 K) while in figure 5d and 5e it is the crystal temperature (460K-500K).

Response:

Thank you for your comments.

In Fig. 4a, we used the water-cooling temperature. In Fig. 5d and 5e, we used the simulated temperature inside the laser crystal by solving heat transport equation. In the calculations of constant of $C = T_{th} \times P_{th}$, the latter one is more accurate. Therefore, to quantitatively investigate the new temperature degree of freedom, we adopt a “real” temperature inside the laser crystal, which is difficult to measure directly.

This temperature simulation has been a well-established method to calculate the thermal-field distribution in laser crystals, see [ref: *Opt. Commun.* 274, 176-181 (2007); *Laser Physics* 25, 075003 (2015)].

The steady-state thermal-field distribution equation is

$$k_x \frac{\partial^2 T}{\partial x^2} + k_y \frac{\partial^2 T}{\partial y^2} + k_z \frac{\partial^2 T}{\partial z^2} + q(x, y, z) = 0, \quad (1)$$

where k_x, k_y, k_z represent the thermal conductivity of Nd:YVO₄ crystal along $x, y,$ and z directions, respectively. $T(x, y, z)$ is the temperature inside the laser crystal, and $q(x, y, z)$ is the heat generation arising from pumping light per unit volume. The intensity of pumping light obeys the Gaussian function. The heat generation for Nd:YVO₄ crystal on the Cartesian coordinate system can be written as

$$q(x, y, z) = \frac{2Q\alpha}{\pi\omega_p^2} (1 - e^{-\alpha l}) e^{-2[(x-\frac{c}{2})^2 + (y-\frac{b}{2})^2] / \omega_p^2} e^{-\alpha z}, \quad (2)$$

where Q is the total heat load in the crystal due to the quantum defect, α denotes the pump absorption coefficient, ω_p is the beam waist, and l is the crystal length. The Eq. (2) is solved by finite element analysis. The boundary conditions are chosen as the lateral-surface temperature of laser crystal controlled by water-cooling. Then, we can obtain the real thermal-field distribution inside the crystal.

As shown in Fig. 5b, with the cooling temperature maintained at 289 K, the real temperature value inside laser crystal is 461 K under the 7.7 W pump power.

Action:

We added some descriptions for temperature simulations in revision Page 12.

Comment:

4. Is the experimental result $C=3388 \pm 158 \text{ W}\cdot\text{K}$ obtained by fitting the experimental data in Figure 5d of the main text, or by using the theoretical formula with other experiment parameters like optical quality factor? If the answer was the former one, did the author compare it with the theoretical result in the second case?

Response:

Thank you for your comments. At present, the constant C was obtained by fitting the experimental data. In supplementary material Section-VI, we gave a qualitative analysis for $P_{th} = C/T_{th}$, but no quantitative calculations for

the values of constant C . At present, it is quite difficult to calculate the constant C by first-principles method or some basic spectral parameters, *e.g.* spectroscopic quality factor.

However, according to our theoretical analysis, the relation between output power P_{out} and temperature T can be written as

$$P_{out} \propto \alpha P_{pump} - \frac{\beta}{T-T_0}. \quad (3)$$

where α and β represent the intrinsic parameters of lasing system, T_0 is a critical temperature. By fitting the experimental result (Fig. R3), we obtain $\alpha=0.083$, $\beta=5.771 \text{ W}\cdot\text{K}$, $T_0=455.36 \text{ K}$. The fitting curve is well consistent with the experimental results, indicating our laser model is believable and credible.

Figure R3. Crystal temperature dependent 1176 nm laser output power at a fixed pump power 7.7 W.

Comment:

5. Lines 254-256 “with lateral-surface temperature increasing from 293 K to 338 K, the P_{th} decreased from 7.6 W to 6.6 W while saturation output power increased from 1.5 to 1.7 W, which has never been reported to our knowledge” is an ambiguous statement and needs to be revised.

Response:

Thank you for your suggestions. We corrected this sentence to “As shown in Fig. 5a, with lateral-surface temperature increasing from 293 K to 338 K, the P_{th} decreased from 7.6 W to 6.6 W while saturation output power increased from 1.5 to 1.7 W, indicating thermally-activated phonon-pumping process indeed exists in Nd:YVO₄ crystal.”

Comment:

6. It is better to replace the statement “see supplementary material” in the main text (for example, lines 94, 190, 204, and 351) with specific places in supplementary material like “see supplementary material Fig S5”.

Response:

Thanks for your suggestions. We have revised these cases in revision.

line 94 (see Supplementary Materials, Section-VII, Steady-state rate equation for PPCP laser)

line 190 (see Supplementary Materials, Section-V, Selection rules of phonons in electron-phonon coupling process)

line 204 (see Supplementary Materials, Section-V, Selection rules of phonons in electron-phonon coupling process)

line 351 (see Supplementary Materials, Section-IV, Difference between alexandrite laser and Nd:YVO₄ laser)

Comment:

7. The citation should be added in line 303 and line 521 (Fuchrbauer-Ladenburg formula).

Response:

Thanks for your suggestions. We have added the citations in line 303 [refs 34, 35] and line 521 [refs 34, 35].

Comment:

8. Line 173-174 “When we further amplify it in the resonant cavity, such temperature step could possibly become a lasing threshold T_{th} ” is obscure and needs to be revised.

Response:

Thanks for your suggestions. We deleted this sentence “~~When we further amplify it in the resonant cavity, such a temperature step could possibly become a temperature threshold T_{th} .~~”

Comment:

9. In order to show the system stability and laser noise, the time domain signal shown in Figure S27 is insufficient, the noise represented in the frequency domain by using the electronic spectrum analyzer might be better to characterize the intensity and phase noise of the device here, and further verified the high stability the author announced here.

Response:

Thanks for your suggestions.

In Fig. S27, the oscilloscope trace was used to demonstrate our Nd:YVO₄ laser at 1176 nm is a continuous-wave (CW) laser. It was not to show the laser noise.

In order to show the laser stability, we measured the power stability and wavelength stability. As shown in Fig. R4, the fluctuations of output power were less than $\pm 1.5\%$ over a period of 30 minutes. Meanwhile, the laser wavelength maintains at 1176 nm with good stability.

Figure R4. a. The power stability measured at an output power of 1.2 W. b. The wavelength stability measured at an output power of 1.2 W.

In addition, according to the reviewer’s suggestions, we measured the laser noise in the frequency domain using the electronic spectrum analyzer (Fig. R5). Our Nd:YVO₄ laser at 1176 nm is a CW laser, so we can’t find the laser signal from frequency spectrum analyzer.

Figure R5. RF spectra of the CW Nd:YVO₄ laser at 1176 nm.

Action: Fig. S28 was updated with Fig. R4.

Comment:

10. I suggest labeling the variation of the pump power in Figures S14-S17, just like the temperature change elucidated in Figure S30.

Response:

Thanks for your suggestions. We added the pump power values in Fig. S14-S17.

Figure R6. The pump dependent fluorescence and laser spectra under normalized condition, $T=293$ K.

Comment:

11. The text in Figures S32 and S35c is too small.

Response:

Thanks for your suggestions. We revised Fig. S32 and S35c.

Figure R7. (a) Laser spectrum of the actively Q-switched self-Raman laser of Nd:YVO₄ crystal. (b) Laser spectrum of our photon-phonon collaboratively pumped laser.

Figure R8. (a) Schematic of a four-level Nd³⁺ laser system, (b) Temperature-dependent laser performances of Nd:YVO₄ at 1064 nm. (c) Temperature-dependent laser performances of Nd:YVO₄ at 1064 nm.

Reviewer #2

Comment:

The authors demonstrate a lasing phenomenon, which is in general counter to other lasers in the sense that there exists a regime of temperature where the threshold current decreases with increasing temperature. The work is original as far as I can see.

However, the explanations as to the exactly mechanisms of the counterintuitive phenomenon are rather inadequate, in part because of 1) the structure of presentations and 2) language issues. But, setting these stylistic problems aside, I am not certain if the phenomenon is mainly because of increased emission cross-section at higher temperatures, or due to phonon pumping as the title and conclusions suggest. And these two things (cross-section and phonon pumping) seem to be unrelated to each other.

Response:

We thank the reviewer for his/her constructive comments.

(1) First, we improved the structure of presentations and English language in the revision. In order to make it clear, we added some discussions on “phonon pumping”, especially on the population inversion of high levels induced by phonon absorption.

(2) The increased emission cross-sections at higher temperatures are related to phonon-pumping process.

As shown in Fig. R8, in a Nd:YVO₄ crystal, the main fluorescence emission locates at 1064 nm, corresponding to a pure-electronic transition from ⁴F_{3/2} to ⁴I_{11/2} level. It is a phonon-free transition, where its fluorescence intensity decreases and fluorescence bandwidth increases at high temperatures.

At high temperatures, the emission cross-section in 1140-1240 nm become strong owing to thermally-induced population fluctuation. However, only for those symmetry-allowed emission, phonon-pumped lasing becomes possible. For example, emission at 1176 nm originates from a new phonon-pumped transition, that the *ions on ⁴F_{3/2} level first move to a higher coupled state by an A_{1g} phonon annihilation*, and then jump to ⁴I_{11/2} level (also coupled state). This transition contains a thermally-activated phonon-pumping process. At high temperatures, the thermal excitation rates for the ①→② (also ①→③) transition increase, thus providing enhanced gain cross-section on the ②→④ laser transitions.

Figure R9. Configuration coordinate diagram of Nd:YVO₄ laser.

Action:

The revised version emphasizes the phonon-pumping mechanism (main text Page 3).

Comment:

In the bulk gain medium, phonon modes have a continue spectrum. Out of all the phonons, what is the percentage of those special phonons that satisfy the symmetric condition for enhanced cross-section?

Response:

We thank the reviewer's comments. YVO₄ crystal belongs to tetragonal I4₁/amd space group, where the Y³⁺ (doped Nd³⁺) ions in the D_{2d} site reside between (VO₄) tetrahedral units. There are 24 atoms in a YVO₄ unit cell. Therefore, the irreducible representation of phonons for YVO₄ crystal can be written as:

$$\Gamma = 10E_u + 4A_{2u} + 36E + B_{2u} + 10E_g + A_{2g} + 4B_{2g} + B_{1g} + A_{1u} + 2A_{1g} + 2B_{1u}$$

According to our selection rules, only A_{1g}, B_{1g}, B_{2g} and E_g modes can contribute to the phonon-pumping process.

Therefore, the percentage of special phonons that satisfy the symmetric condition for enhanced cross section is $\frac{17}{72}$.

Among them, two phonons, A_{1g} mode at 891 cm⁻¹ and E_g mode at 839 cm⁻¹, represent the strongest coupling modes in phonon-pumping process, corresponding to laser wavelengths at 1176 nm and 1168 nm, respectively. These two laser wavelengths have been demonstrated in our experiments.

Action:

We added these discussions in supplementary materials, Section-V.

Comment:

Phonon pumping does not seem to increase the population inversion and the total number of ions in the upper lasing state, which is determined by the optical pumping level (phonon only move them to higher energies), how can the output laser power increase with temperature? If the existence of phonons does not increase the number of ions in the upper lasing state, how can you call this phonon pumping? Also, the definition of threshold generally is related to achieving a certain level of population or population inversion. The use of the term "temperature threshold" does not appear to be appropriate if indeed the phonons do not increase population/population inversion.

Response:

Thanks for your comments.

We agree that “phonon move upper ions to higher energies”. In this process, there exists phonon absorption in up-moving of active ions (similar to photon absorption in optical pumping). We also agree that “The definition of threshold generally is related to achieving a certain level of population or population inversion”. As shown in Fig. R9, the population at ② state is related to phonon-pumping step from ① to ②. This is a thermally-activated process depending on temperature, so “temperature threshold” is appropriate to describe the emergence of this physical process.

Here, we want to emphasize the whole lasing process in our model. In conventional Nd-lasers, ①→⑤ transition is natural at a finite temperature. It represents a phonon-free laser at 1064 nm. However, if we suppress the ①→⑤ laser oscillation by specific coating cavity, and amplify another transition channel ②→④ at 1176 nm, photon-phonon collaboratively pumped lasing become available. At this time, the population of upper level ② is determined by two parts, that inherent population ① from photon-pumping and thermally-activated phonon-pumping from ① to ②. Clearly, a population inversion happens between ② and ④ states ($n_② > n_④$ for lasing), when there are enough active ions at ① state and sufficient thermal phonons to support ①→② moving.

The pumping rate for ①→② is determined by the temperature ($W_{①→②} \propto T$), because the phonon occupation numbers n_p are given by the Bose-Einstein distribution function,

$$n_p = \frac{1}{e^{\hbar w_p/k_B T} - 1}$$

where \hbar is Planck constant, w_p is phonon frequency, k_B is Boltzmann's constant, and T is the temperature. A higher T , a larger n_p will be. If temperature is too low (small $W_{①→②}$), the population of ② would not be large enough to support population inversion ($n_② < n_④$), and the lasing at 1176 nm is not allowed. At high temperatures, the thermal excitation rates for ①→② transition increase, thus providing increased emission cross-section on the ②→④ laser transitions. Therefore, the population inversion for 1176 nm laser is strongly related to the phonon density of states, as well as crystal temperature.

Action:

We added the discussion for population inversion Δn in main text Page 4 and Supplementary Materials Section-VII.

“As shown in Fig. 1a, the population of lasing up-level $n_{1''}$ is

$$n_{1''} = \frac{W_p}{e^{\frac{E_{1''} - E_{0''}}{k_B T}} (W_{0''2''} + A_{0''2''} + 2S_{0''1''}) - (W_{1''2''} + A_{1''2''})}, \quad (1)$$

and the population of lasing low-level n_2 is

$$n_2 = \frac{W_p \left[e^{\frac{E_1' - E_0'}{k_B T}} (W_{0'2'} + A_{0'2'}) + (W_{1'2'} + A_{1'2'}) \right]}{S_{2'0} \left[e^{\frac{E_1' - E_0'}{k_B T}} (W_{0'2'} + A_{0'2'} + 2S_{0'1'}) - (W_{1'2'} + A_{1'2'}) \right]}, \quad (2)$$

As a result, the population inversion Δn can be written as equation (3)

$$\Delta n = n_1 - n_2 = \frac{W_p \left[S_{2'0} - e^{\frac{E_1' - E_0'}{k_B T}} (W_{0'2'} + A_{0'2'}) - (W_{1'2'} + A_{1'2'}) \right]}{\left[e^{\frac{E_1' - E_0'}{k_B T}} (2S_{0'1'} + W_{0'2'} + A_{0'2'}) - (W_{1'2'} + A_{1'2'}) \right]} S_{2'0}, \quad (3)$$

where n_i represents the electron population densities of level i , W_{ij} is the transition probability for stimulated radiation between level i and j , A_{ij} is the spontaneous transition probability, and S_{ij} is the nonradiative transition probability, W_p is optically pump rate, $E_1' - E_0'$ is the energy separation between level 1' and 0', k_B is Boltzmann's constant, T is temperature. It is seen that higher temperature T is helpful for the formation of population inversion. If the temperature is sufficiently high, a two-phonon pumping process can also be expected, corresponding to the upper level of 2' state. Therefore, there is actually a synergetic photon-phonon pumping process. In a circle, electron return back to the ground state with two non-radiative transition processes, $2' \rightarrow 0'$ and $LS \rightarrow GS$ (${}^4F_{11/2} \rightarrow {}^4I_{9/2}$ in Nd^{3+} -crystals), associated with thermal phonon emission and heat creation.

Comment:

Lastly, in view of the comment above regarding percentage of phonons that satisfy symmetric requirement, what are the power efficiency (accounting for both optical pump power and power used to heat the crystal), in comparison with pure optical pumping? If there is a penalty in efficiency, what can be the possible applications of this type of lasers.

Response:

Thank you for your suggestion. A comparison for laser efficiency was listed in Table R1.

Table R1. A comparison between 1064 nm laser and 1176 nm laser in Nd:YVO₄ crystal.

Nd:YVO ₄ laser	pump wavelength (nm)	theoretical efficiency $\eta_{\text{cal}} = \lambda_p / \lambda_L$	experimental slope efficiency η_{exp}	$\eta_{\text{exp}} / \eta_{\text{cal}}$	reference
phonon-free laser at 1064 nm	808	76%	58.6%	77.1%	Laser Phys. Lett. 7, 210-212 (2010)
PPCP laser at 1176 nm	808	69%	34.3%	49.7%	this work
self-Raman laser at 1176 nm	808	69%	21%	30.4%	Chin. Phys. B. 25, 114207 (2016)

The energy of incident pump light is mainly used to excite the Nd^{3+} ions from ${}^4I_{9/2}$ to ${}^4F_{5/2}$ level. The heat generation originates from nonradiative transitions of ${}^4F_{5/2} \rightarrow {}^4F_{3/2}$ and ${}^4I_{11/2} \rightarrow {}^4I_{9/2}$. For phonon-free 1064 nm laser and phonon-involved 1176 nm pumped by 808 nm InGaAs laser diode, the quantum defect is 24% and 31%, respectively. As a result, the theoretical-limit efficiencies of them are comparable (76% and 69%). In previous reports, the highest slope

efficiency for 1064 nm laser is 58.6%. In contrast, our Nd:YVO₄ laser at 1176 nm exhibit a high slope efficiency of 34.3%. Therefore, our PPCP laser doesn't exist a penalty in efficiency, but a comparable efficiency to phonon-free 1064 nm laser.

In addition, our PPCP laser also exhibits a higher laser efficiency than conventional self-Raman laser at 1176 nm. To the best of our knowledge, the highest slope efficiency of Nd:YVO₄ self-Raman laser is 21%, lower than our laser. Therefore, our PPCP laser at new wavelengths has a sufficient conversion efficiency for practical applications.

Applications:

This phonon-pumping mechanism can be feasible in many laser materials and bring great opportunities for wavelength extension and power improvement at high temperatures. For example, we have realized phonon-pumped lasers at 1151 and 1166 nm in Nd:YAG crystal. These two wavelengths are never reported in Nd:YAG crystal. In addition, this new lasing mechanism is controllable by coating cavity, suggesting its great applications in solid-state laser engineering.

Action:

In the revision Page 14, we added “*This new collaborative lasing mechanism is promising for searching new-wavelength lasers via phonon symmetry design or multi-phonon coupling processes. For example, we apply this PPCP mechanism in Nd:YAG crystal and obtain some unconventional lasers at 1151 and 1166 nm.*”

Reviewer #3

Comment:

Manuscript Report: "Photon-phonon collaboratively pumped laser"

In this manuscript, the authors report on the experimental demonstration of a phonon-assisted (thermally-induced) laser at 1176nm wavelength. The authors demonstrate a novel laser system that is collectively pumped by both photons and phonons. The authors provide convincing experimental evidence showing conventional power thresholds and input-output power corresponding to a Nd:YVO₄ laser at 1176 nm wavelength. The developed laser is characterized by an unusual temperature lasing threshold: a critical temperature where the output laser increases as the temperature increases while the pump power is sustained. Furthermore, the authors provide a detailed analysis of the temperature dependence which will be very useful to implement laser systems using the different transitions of Nd:YVO₄. To the best of my knowledge, this laser mechanism and the enhanced performance with temperature has never been observed in lasers based on Nd materials. Hence, the novelty of the present work unquestionable and, in my opinion, this work is a valuable contribution for the laser physics community.

I only have a minor concern, I noticed that the authors have published a paper on a similar topic, "Multiphonon-assisted lasing beyond the fluorescence spectrum", in Nature Physics 18, 1312-1316 (2022), and I am wondering how is that work related to the present one? In other words, if both papers deal with phonon-assisted lasers why the nature physics paper is not even cited in the present manuscript?

After clarification of this minor concern, I will be in position to recommend the present manuscript for publication in Nature Communications.

Response:

Thank you for your positive comments.

We first submitted this Nd:YVO₄ work to *Nature Photonics* in 18th-May-2022, and then transferred to *Nature Communications* in 5th-July-2023. At the first submission, our paper in nature physics has not been published. So, we missed this citation. Now, we added this important citation in the revision, see ref [20].

Difference between Nd:YVO₄ laser and Yb:YCOB laser in Nature Physics 18, 1312-1316 (2022).

Nd:YVO₄ laser and Yb:YCOB laser hold different physical mechanism. The former one is the thermally-activated phonon-pumping process with absorbing one or two phonons, whereas the latter one exhibits the multiphonon-assisted laser wavelength extension with emitting three- to eight-phonons.

In previous work in nature physics, we realized a broadband tunable laser generation far beyond the fluorescence spectrum in Yb:YCOB crystal. Yb:YCOB is a strong electron-phonon coupling crystal with Huang-Rhys S factor of 1.34. The main highlight is the great wavelength tuneability and extension by multiphonon-assisted lasing process. We obtained an overall spectral tuneability of 1110-1465 nm in five segments, each with its own configuration, corresponding to the three- to seven-phonon cases. The longest (eight-phonon) lasing wavelength can reach 1518 nm, over 400 nm beyond the fluorescence spectrum.

In this work, Nd:YVO₄ is a weak-coupling system with Huang-Rhys S factor of 0.013. Despite of such small coupling intensity, electron-phonon coupling is also important for phonon-involved laser. Here, we focus on the temperature-dependent laser performances of Nd:YVO₄ crystal. The main highlight is an anomalous phonon-pumped lasing phenomenon in Nd-laser materials, which is counter to other lasers in the sense that there exists a regime of

temperature where the lasing threshold decreases with increasing temperature. A phonon-photon collaboratively pumped laser with a lower lasing power threshold P_{th} at higher temperature T_{th} has been reported ($P_{th} \times T_{th} = \text{constant}$). A critical temperature threshold T_c was verified experimentally to claim the phonon-pumping mechanism.

A list of changes:**Text**

We slightly revised the abstract and introduction parts.

Several sentences have been added to address the reviewers' comments, especially for phonon-pumped lasing mechanism, and laser wavelengths stability.

References

References have been revised in the main text.

Supplementary information

Additional extended Fig. S1, S28, Table S1, and explanatory text have been updated in the supplementary materials.

REVIEWER COMMENTS

Reviewer #1 (Remarks to the Author):

The response of this manuscript has addressed most of my concerns in the initial review, and without a doubt, the quality of this manuscript has improved in this revision. However, I still find the comparison between Nd:YVO4 in this work and the alexandrite laser (Rulin Miao, Yu Fu, Dazhi Lu, et al, Deciphering the vibronic lasing performances in an electron-phonon-photon coupling system. Opt. Express 31, 9790-9799 (2023).) somewhat confusing. Therefore, I suggest the author provide further clarification regarding the innovation of this work before final publication.

Please see the attachment for the details.

Reviewer #2 (Remarks to the Author):

The authors mostly addressed my questions using the same material in the original manuscript so I still have issues.

Regarding the relationship between emission cross section and phonon pumping, the answer is still not satisfactory. The 4 levels 1,2, 3, 4 are not clearly indicated in the manuscript or supplement. Perhaps in this particular case, the laser emission cross section and the pump absorption cross section are not equal, which is why it is hard to understand the statement "The increased emission cross-sections at higher temperatures are related to phonon-pumping process". I think threshold is more related to pump absorption cross-section rather than the emission cross section. The term gain cross section further confuses the reviewer.

As to the power efficiency, the author listed only 1 other laser for comparison, which is rather limited. Semiconductor lasers at that wavelength may have much higher wall-plug efficiencies. Are there any other solid state lasers that have better efficiency? So, the author need to give a much more thorough explanations.

Reviewer #3 (Remarks to the Author):

The authors have addressed my concern and other's reviewers comments from the previous review round. Now, I am in position to recommend this work in Nature Communication.

The response of this manuscript has addressed most of my concerns in the initial review, and without a doubt, the quality of this manuscript has improved in this revision. However, I still find the comparison between Nd:YVO4 in this work and the alexandrite laser (Rulin Miao, Yu Fu, Dazhi Lu, et al, Deciphering the vibronic lasing performances in an electron-phonon-photon coupling system. Opt. Express 31, 9790-9799 (2023).) somewhat confusing. Therefore, I suggest the author provide further clarification regarding the innovation of this work before final publication

1. In this work, the authors have stated that the gain can be expressed $G = n_{21}\sigma$ (equation S7), and the emission cross section σ scales with the temperature T . Consequently, the gain G scales with T , and the lasing threshold power P_{th} is inversely proportional to the temperature threshold T_{th} . However, based on the result of the steady-state rate equation in supplementary materials **Section VII**, the population inversion n_{21} is also influenced by the temperature T (equation 1, main text), **as a result, it seems the theoretical conclusion $P_{th} \times T_{th} = \text{constant}$ is not entirely convincing.**

2. Regarding the $\text{Cr}^{3+}:\text{BeAl}_2\text{O}_4$ alexandrite laser in Opt. Express 31, 9790-9799 (2023), the paper illustrates that the variation of temperature affects the emission cross section σ and finally $P_{th} \propto 1/F_{em}(\lambda, T)$, where F_{em} is the lineshape functions of emission and absorption spectra which is influenced by temperature T :

$$F(\lambda, T) = |M_{ij}|^2 e^{-s(2n+1)} \sum_V \frac{S(n+1)^{v+p} (Sn)^v}{(v+p)! v!} \quad (\text{Eq 001})$$

Where $S(T) = \frac{1}{N} \sum_s \left(\frac{\omega_0}{2h}\right) \Delta_{jis}^2$. By examining equations S9-S12 in this article, it is apparent that this work represents a special case of equation R1 ($S \rightarrow 0$), and the Huang-Rhys factor S (2.13) in Opt. Express 31, 9790-9799 (2023) is simply larger.

Furthermore, in the “**Response to reviewer’s comments**”, the author asserts that two close 4T_2 and 2E upper levels are generated by crystal-field Stark splitting in alexandrite laser is the core reason for the decreasing of the lasing threshold power P_{th} . However, based on the provided equation, it’s challenging to discern the importance of this effect from the quantitative perspective of view, the mechanism behind the alexandrite laser and Nd:YVO4 seems quite similar when analyzing the form of

Equation 001.

Additionally, in the supplementary material, **Section-IV** “In the alexandrite laser, the terminal laser level is a set of vibrational states well above the ground state. The laser’s initial level is a level 800 cm^{-1} above a long-lived storage level and in thermal equilibrium with it. The laser terminal level is very close to the ground-state level. Therefore, raising the temperature also tends to populate the terminal level...In comparison, the laser low-level of Nd:YVO₄ is ${}^4\text{I}_{11/2}$ level, not ground-state level ${}^4\text{I}_{9/2}$. There is a large energy difference of 1966 cm^{-1} . In general, the population of ${}^4\text{I}_{11/2}$ level can be neglected in the calculation of steady-state population inversion” The author claimed that the V-shape temperature-dependent performance shows up due to the terminal level population for the alexandrite laser, while the terminal level population can be neglected for Nd:YVO₄ laser of this study. **Does this imply that the V-shaped temperature-dependent performance can also be observed for Nd:YVO₄ laser as long as the environment temperature is sufficiently high?** (it’s worth noting that the range of temperature tuning in this experiment appears to be relatively narrow, as indicated in Figure R2, and the V-shaped temperature-dependent performance was not reported for the alexandrite laser in Opt. Express 31, 9790-9799 (2023), either.)

Response to reviewer's comments

Reviewer #1

Comment:

The response of this manuscript has addressed most of my concerns in the initial review, and without a doubt, the quality of this manuscript has improved in this revision. However, I still find the comparison between Nd:YVO₄ in this work and the alexandrite laser (Rulin Miao, Yu Fu, Dazhi Lu, et al, Deciphering the vibronic lasing performances in an electron-phonon-photon coupling system. *Opt. Express* 31, 9790-9799 (2023).) somewhat confusing. Therefore, I suggest the author provide further clarification regarding the innovation of this work before final publication.

Response:

Thank you for your comments. Alexandrite laser is a typical three-level system with strong electron-phonon coupling effect, while Nd:YVO₄ is a four-level system with weak electron-phonon coupling effect. This is the key difference for these two laser crystal.

In addition, the differences between Nd:YVO₄ laser and alexandrite laser can be summarized in three aspects:

(1) The value of Huang-Rhys factor and the role of phonon.

Alexandrite crystal has a large S factor, and the thermal phonons participated in the lasing process play an average effect for homogeneously spectral broadening [$S \sim 2.13$ & $h\omega_{ph} \sim 524 \text{ cm}^{-1}$ in our paper *Opt. Express* 31, 9790 (2023)].

In comparison, Nd:YVO₄ crystal has a tiny S factor (0.002-0.013). Only specific-symmetry phonons that follow selection rules, such as A_{1g} or E_g phonon mode coupled to $\Gamma_6 \rightarrow \Gamma_7$ transition, are amplified to create quantized laser upper-levels.

(2) Different lasing mechanism.

Alexandrite laser is a particular case in Cr³⁺-doped laser materials, possessing distinctive electronic levels where two close 4T_2 and 2E upper levels are generated by crystal-field Stark splitting (Fig. R1a). Benefitting from a long lifetime of 2E level, thermal excitation rates for the ${}^2E \rightarrow {}^4T_2$ transition increase at high temperature, thus providing enhanced gain on the ${}^4T_2 \rightarrow {}^4A_2$ laser transitions.

For Nd:YVO₄ laser, there is no 2E -like storage level with long lifetime (Fig. R1b). The newly created laser wavelengths associated with the upper levels owe entirely to the enhancement of electron-phonon coupling on the cost of phonons.

Figure R1. Different lasing mechanisms. (a) Alexandrite laser Cr³⁺:BeAl₂O₄, (b) Nd:YVO₄ laser.

(3) Different temperature-dependent relation.

In alexandrite lasers, high temperature also tends to populate the terminal 4A_2 levels. Therefore, the laser performance is the best one with a maximally populated initial level and a minimally terminal one at an optimal temperature (T_m). The typical temperature-dependent performance is V-shape [see *Opt. Lett.* 7, 608 (1982), $T_m \sim 225^\circ\text{C}$].

In Nd:YVO₄ laser, high temperature brings more phonons, which will continuously improve laser performances in principle. Most importantly, we quantitatively obtain a lasing threshold-temperature relation, $P_{th} \times T_{th} = \text{constant}$, based on both theoretical analysis and experimental results.

In summary, we make a comparison between alexandrite and Nd:YVO₄ laser in Table R1. These two lasers are different in laser mechanism, temperature dependence, and laser performances. Accordingly, our Nd:YVO₄ work is original and full of novelty.

Table R1. A comparison between alexandrite and Nd:YVO₄ laser.

Crystal	pump wavelength	ΔE (cm ⁻¹)	phonon energy (cm ⁻¹)	laser wavelength λ (nm)	pump threshold P_{th} (W)	temp threshold T_{th} (K)	$P_{th} \times T_{th} = C$	C (W·K)
Cr ³⁺ :BeAl ₂ O ₄	638 nm	$\Delta E = {}^4T_2 - {}^2E = 800 \text{ cm}^{-1}$	average phonon $h\omega_{ph} = 524 \text{ cm}^{-1}$	761-772	3 ~ 4 W	No	No	---
Nd ³⁺ :YVO ₄	808nm	determined by the selection-allowed phonons	$A_{1g} = 891 \text{ cm}^{-1}$	1176	6 ~ 8 W	293-308 K	Yes	3388±158

Reviewer #2

Comment:

The authors mostly addressed my questions using the same material in the original manuscript so I still have issues. Regarding the relationship between emission cross section and phonon pumping, the answer is still not satisfactory.

The 4 levels 1, 2, 3, 4 are not clearly indicated in the manuscript or supplement. Perhaps in this particular case, the laser emission cross section and the pump absorption cross section are not equal, which is why it is hard to understand the statement "The increased emission cross-sections at higher temperatures are related to phonon-pumping process". I think threshold is more related to pump absorption cross-section rather than the emission cross section. The term gain cross section further confuses the reviewer.

Response:

We thank the reviewer for his/her constructive comments.

(1) First, we updated Fig. 1 in the revision. The levels ①, ②, ③, ④, ⑤, are clearly indicated in the manuscript.

Figure R2. (a) Configuration coordinate diagram of PPCP laser. (b) 2D temperature-power lasing phase diagram.

(2) Temperature-dependent pump absorption.

We agree that "in this particular case, the laser emission cross section and the pump absorption cross section are not equal". Nd:YVO₄ crystal has no absorption at the wavelength of 1176 nm and 1064 nm, as described by the four-level systems as shown in Fig. R2. Meanwhile, there is the maximum emission at 1064 nm and an absorption peak at 808 nm. They are not equal.

In addition, we measured the temperature-dependent pump absorption in our lab. As shown in Fig. R3, the absorbed pump power gradually decreased with the increasing temperatures. The absorption efficiency reduced from 66.9% at 283 K to 61.3% at 353 K. This case is well consistent with previous report, where the absorption cross section of Nd:YVO₄ decreased from $(58.6 \pm 0.2) \text{ pm}^2$ at 291 K to $(30.9 \pm 0.6) \text{ pm}^2$ at 430 K [ref: Silvia Cante, University of Southampton, (2021), PhD thesis, Page 57].

Figure R3. (a) Absorbed pump power versus incident pump power under 283-353 K. (b) Temperature-dependent absorption cross-section of Nd:YVO₄ crystal, adapted from [Silvia Cante, University of Southampton, PhD thesis (2021)].

More importantly, the pump power in main-text Fig. 5a refers to the absorbed pump power in experiments. Based on the well-developed theory of diode-pumped solid-state lasers (DPSSLs) [ref: IEEE J. Quantum Electron. 24, 895, 1988], the absorbed pump power $P_{a,th}$ at threshold can be described as Eq. (1),

$$P_{a,th} = \frac{\pi h \nu_p}{2 \sigma_e \eta_p \tau} (\omega_0^2 + \omega_p^2) \left(\frac{\delta_e}{2} + \alpha_i L + \alpha_l L \right) \quad (1)$$

where $h \nu_p$ is the pump photon energy, σ_e is the effective stimulated emission cross-section, η_p is the pump quantum efficiency which is the number of ions in the upper manifold created by one absorbed photon, τ is the upper manifold lifetime, ω_0 is the Gaussian beam radius of the laser cavity mode, ω_p is the Gaussian beam radius of the pump beam, δ_e is the extrinsic loss (scattering at interfaces, and Fresnel reflections), $\alpha_i L$ represents losses which are proportional to the gain medium length such as impurity absorption, α_l is the absorption coefficient at the laser wavelength in the gain medium due to the lower laser level population, L is the gain medium length. From Eq. (1), we can see that the lasing threshold $P_{a,th}$ on the absorbed pump power does not depend on the pump absorption coefficient under various temperatures.

Therefore, the low-threshold of Nd:YVO₄ 1176 nm laser at high temperature cannot be attributed to the increased pump absorption cross-section. In our opinion, the increased emission cross-section σ_e at 1176 nm is more important for the reduction of lasing threshold, which is related to the enhanced phonon-pumping at high temperatures.

(3) “gain cross-section” was corrected to “emission cross-section”. We are sorry for this typo and confusion to reviewers.

Action:

1. In revision, we updated Figure 1 in the manuscript with level-①, ②, ③, ④, ⑤. In Fig. 1 Caption, we added “In conventional Nd³⁺-lasers, ①→⑤ transition at 1064 nm is natural at a finite temperature. It represents a phonon-free laser oscillation. If we suppress the ①→⑤ laser oscillation by specific coating cavity, and amplify another transition channel ②→④, photon-phonon collaboratively pumped lasing at 1176 nm become available. At this time, a population inversion happens between ② and ④ states ($n_2 > n_4$ for lasing), when there are enough active ions at ① state and sufficient thermal phonons to support ①→② up-moving.”
2. In main-text Page 18, we added “The pump power refers absorbed value through the whole paper.”
3. In Supplementary Materials (Section-II), we added Fig. R3 as Fig. S35 and the related discussion.

4. In main-text Page 19, we added “The absorbed pump power gradually decreased with the increasing temperatures, indicating the low-threshold of 1176 nm laser at high temperature cannot be attributed to the increased pump absorption cross-section.”

Comment:

As to the power efficiency, the author listed only 1 other laser for comparison, which is rather limited. Semiconductor lasers at that wavelength may have much higher wall-plug efficiencies. Are there any other solid-state lasers that have better efficiency? So, the authors need to give a much more thorough explanation.

Response:

Thank you for your suggestion. Here, we give a comprehensive comparison for conversion efficiency of many solid-state lasers, including *diode-pumped solid-state lasers, optically-pumped Raman lasers, optically-pumped semiconductor lasers, electrically-pumped semiconductor lasers, and our PPCP laser.*

Table R2. A comparison for solid-state lasers and semiconductor lasers.

type	laser	pump wavelength λ_p (nm)	temporal characteristics	slope efficiency	optical-to-optical efficiency	reference
PPCP laser	Nd:YVO ₄ PPCP laser at 1176 nm	808	cw	34.3%	17.7%	this work
LD-pumped self-Raman laser	Nd:YVO ₄ self-Raman laser at 1176 nm	808	cw	8.1%	7.7%	Chin. Phys. Lett. 28, 054202 (2011)
		808	pulse (ns)	19.2%	13.9%	Opt. Lett. 29, 1915-1917 (2004)
		808	pulse (ns)	--	18.2%	Laser Phys. Lett. 6, 26-29 (2009)
		808	pulse (ps)	8.2%	5.3%	Opt. Laser Technol. 89, 1-5 (2017)
		878.9	cw	21%	20%	Chin. Phys. B. 25, 114207 (2016)
		880	cw	8.98%	7.3%	Opt. Lett. 40, 3524-3527 (2015)
		878.6	quasi-cw	--	28.6%	Infrared and Laser Engineering 50, 20200227 (2021)
LD-pumped self-Raman laser	Nd:YVO ₄ +YVO ₄ laser at 1176 nm	808	cw	8.5%	7.8%	Appl. Phys. B 103, 559-562 (2010)
	YVO ₄ +Nd:YVO ₄ +YVO ₄ laser at 1176 nm	808	cw	14.6%	13.3%	Acta. Phys. Sin. 63, 154208 (2014)
	YVO ₄ +Nd:YVO ₄ +YVO ₄ laser at 1176 nm	808	ns	16.3%	12.4%	Appl. Phys. B 101, 743-746 (2010)
Optically-pumped Raman laser	Nd:YAG+YVO ₄ Raman laser at 1176 nm	808	pulse (ns)	--	8.8%	J. Phys. D: Appl. Phys. 50, 465303 (2017)
	diamond Raman laser at 1178 nm	1018	cw	38%	24.4%	Appl. Phys. Lett. 121, 141104 (2022)
Optically-pumped solid-state laser	Nd:YAG-SHG pumped Ti:sapphire laser at 1178 nm	532	pulse (ns)	--	0.48%	Conference on Lasers and Electro-Optics. 14, 296-298 (1987)
	diode-pumped Yb:LCB laser at 1162 nm	976	cw	8.4%	7%	Light Sci. & Appl. 12, 203 (2023)
	diode-pumped Yb:YCOB laser at 1200 nm	976	cw	16.4%	15%	Opt. Lett. 48, 4913-4916 (2023)
Optically-pumped semiconductor laser	VECSEL laser at 1180 nm	808	cw	33%	27.3%	Electronics Letters 49, 59-60 (2013)
	VECSEL laser at 1178 nm	808	cw	29.4%	27%	Proc. SPIE 9349, 93490U (2015)
	VECSEL laser at 1185.5 nm	808	cw	38%	29%	Electronics Letters 54, 1135-1137 (2018)
Electrically - pumped semiconductor laser	semiconductor DBR laser at 1180 nm	--	cw	--	electrical-to-optical efficiency = 31%	Opt. Lett. 41, 657-660 (2016)

As listed in Table R2, among LD-pumped solid-state lasers at 1176 nm, our PPCP laser exhibits the highest slope efficiency at 1176 nm. In addition, compared to the state-of-art diamond-Raman laser at 1178 nm, our PPCP laser also has a comparable slope efficiency [38% (diamond-Raman laser) v.s. 34.3% (PPCP laser)], and a comparable optical-to-optical conversion efficiency [24.4% (diamond-Raman laser) v.s. 17.7% (PPCP laser)].

Moreover, our laser efficiency is also comparable to optically-pumped (or electrically-pumped) semiconductor laser at 1180 nm. Therefore, our PPCP laser doesn't exist a penalty in efficiency, but a comparable (even better) efficiency to other solid-state lasers.

Action:

1. In the revision, we added *Table S6* in the Supplementary Materials.
2. Main text Page 14, “This high efficiency is comparable to diode-pumped self-Raman lasers, optically-pumped Raman lasers, as well as optically-pumped semiconductor lasers (see Supplementary Materials, Section-VIII)”

Reviewer #3

Comment:

The authors have addressed my concern and other's reviewers comments from the previous review round. Now, I am in position to recommend this work in Nature Communication.

Response:

We sincerely appreciate the reviewer's positive comments and recommendation.

A list of changes:

Text

We slightly revised the main text. Several sentences have been added to address the reviewers' comments, especially for phonon-pumped lasing mechanism, and laser efficiency.

Supplementary information

Table S6, Fig. S35, and explanatory text have been updated in the Supplementary Materials.

REVIEWERS' COMMENTS

Reviewer #2 (Remarks to the Author):

The authors have addressed my concerns in a satisfactory manner.